# Understanding pre-training data effects in retinal foundation models using two large fundus cohorts

Yukun Zhou [1,2,3,4,20] ✉, Zheyuan Wang[1,5,6,20], Yilan Wu [1,7,20], Ariel Yuhan Ong [1,2], Siegfried K. Wagner [1,2], Eden Ruffell [1,2,3], Mark A. Chia[1,8], Zhouyu Guan [9], Lie Ju[1,2], Justin Engelmann [1,2], David A. Merle[1,2], Tingyao Li [5,6], Jia Shu[5,6], Paul Nderitu[1,2], Ke Zou[10], Jocelyn Hui Lin Goh[10,11,12], Qingshan Hou[10], Xiaoxuan Liu [13,14], Yaxing Wang[7,15], Yih Chung Tham [10,11,12], Andre Altmann [3,16], Carol Y. Cheung [17], Daniel C. Alexander [3,4], Eric J. Topol [18], Alastair K. Denniston [13,14], Tien Yin Wong [7,12,15,19], Bin Sheng [5,6] ✉ & Pearse A. Keane [1,2] ✉

Medical foundation models, pre-trained on large-scale unlabelled data, show strong performance and data efficiency when adapted to various clinically relevant applications. However, how pre-training data shape the generalisability and fairness of these models remains unexplored. Here we address this using two cohorts from Moorfields Eye Hospital (UK) and the Shanghai Diabetes Prevention Program (China), each containing 904,170 fundus photographs for model pre-training. Using identical pipelines, we train parallel foundation models using individual cohort and evaluate them on downstream tasks with publicly available datasets and held-out data from each site. The parallel models show competitive performance to data that differ substantially from their pre-training data. Nevertheless, we observe fairness gaps over age subgroups, whereas sex and ethnicity show minimal impact. These results demonstrate the good generalisability of retinal foundation models and indicate that pre-training demographic attributes shape fairness differently, highlighting the importance of domain-specific, fine-grained data curation for efficient foundation model development.

Foundation models (FM) are large artificial intelligence (AI) models trained using data and computation at scale[1,2]. Using self-supervised or unsupervised learning methods, FMs capture abundant data patterns that can potentially be applied to diverse applications in real-world scenarios. This approach has broad applications across fields of medical AI[3–5], such as ophthalmology[6–8], radiology[9–11], pathology[12–16], and as a generalist[17,18] advancing clinically meaningful tasks like disease diagnosis and prognosis. However, despite the increasing number of medical FMs being developed, there remains very limited knowledge regarding how the composition of pre-training data, the "guts" of these

models, affects FM capabilities such as generalisability and fairness. The lack of this critical knowledge makes pre-training data collection and medical FM development inefficient and highly speculative.

Training data is the fundamental substrate for developing AI models, with its characteristics encompassing attributes, properties, and features that influence data quality, usability, and its impact on analytical processes. These characteristics define the scope of knowledge and largely determine model capability including generalisability and fairness[19–25]. Previous studies have investigated the impact of labelled data on traditional application-specific AI models in

supervised learning[20,23,26–28], providing effective guidance for labelled data selection during model training. For instance, some work has demonstrated that model performance markedly drops when applied to external sites with distinct data characteristics such as demographics, imaging devices, and disease phenotypes[20,23,26,28]. Additionally, imbalanced training data often leads to poor performance in underrepresented subgroups in terms of age, sex, and ethnicity, raising concerns about model fairness and generalisability[27,29]. To address these challenges, previous work has focused on building diverse and balanced training data for traditional application-specific AI models, using techniques such as data augmentation and synthesis [20,30–32].

Unlike application-specific AI models that rely solely on labelled data for training, FMs learn generalisable features through extensive pre-training on substantial unlabelled data (e.g. via self-supervised learning[33–35]), followed by fine-tuning on labelled data for specific target applications. The performance of FMs is collectively shaped, where pre-training establishes the foundational capabilities while fine-tuning refines performance in specific tasks. Prior studies have systematically shown that the distribution of fine-tuning data, particularly demographic attributes such as age and sex, can be encoded by AI models and lead to fairness gaps[23,25]. However, the impact of unlabelled pre-training data on medical FMs remains underexplored, partly hampered by the substantial workload and resources required to build and compare parallel FMs on matched large-scale datasets from different countries and areas. This lack of knowledge leaves critical questions unanswered: 1) Do pre-training data characteristics affect the generalisability of medical FMs, such as performing poorly on sites with distinct demographics and imaging devices? 2) Do demographic attribute of pre-training data impact medical FM fairness over age, sex, and ethnicity, similar to traditional application-specific AI models? Addressing these questions is critical for understanding how pre-training impacts medical FMs, providing evidence for efficient data sampling to optimise FM generalisability and fairness. Furthermore, revealing the impact of pre-training data provides a strong basis for advocating data transparency, one of the least transparent dimensions according to Foundation Model Transparency Index Scores[36]. Disclosing how pre-training data impacts FMs, and providing details of pre-training data, are essential to understanding FM strengths and limitations [37].

To address these gaps, we investigated the impact of pre-training data on retinal FMs, one of the pioneering areas in medical FM, using two large data cohorts from Moorfields Eye Hospital (MEH), UK, and Shanghai Diabetes Prevention Program (SDPP), China, each comprising 904,170 retinal colour photographs for FM pre-training. We first characterised data cohorts using demographic and imaging metadata, latent features (representative features encoded by models), and clinically meaningful morphological indices, highlighting the differences between the datasets. We then developed retinal FMs with MEH data (FM-MEH) and SDPP data (FM-SDPP), using identical pre-training strategies and implementation details. We evaluated the performance of parallel FMs across a wide range of downstream tasks, including ocular disease diagnosis and systemic event prediction, using data from each site (held out from pre-training data) and publicly available datasets. We finally assessed model fairness across subgroups based on age, sex, and ethnicity. Our results showed the strong generalisability of retinal FMs. For example, FM-MEH and FM-SDPP performed comparably across all SDPP downstream tasks. However, the models exhibited performance gaps over age subgroups (not seen in sex and ethnicity), likely caused by larger distributional shift across age subgroups. The age fairness improved when we supplemented pre-training data with synthetic images that provided a complementary age distribution. Through extensive experiments with real-world clinical data, this study examined the impact of pre-training data on retinal FMs. While our study focuses on retinal images, the observed combination of strong generalisability and uneven demographic effects may extend to broader medical FMs. These findings suggest an evidence-based approach to data description and selection in FM development to improve both efficiency and capabilities.

## Results

### Distinct data distribution in two cohorts

Figure 1 provides an overview of the development and application of retinal FMs. FM-MEH and FM-SDPP were constructed using data from MEH and SDPP, respectively. We randomly sampled 904,170 retinal fundus photographs from each database for FM pre-training, and analysed their characteristics across demographic and imaging metadata, latent features, and clinically meaningful morphological indices. As shown in Fig. 2, significant differences were observed between the MEH and SDPP data in terms of demographic and imaging metadata. The average age in the MEH cohort is 68.88 years (95% Confidence Interval (CI) 68.85, 68.91), significantly older than the SDPP cohort, which had an average age of 47.26 years (95% CI 47.21, 47.30) ($P < 0.001$). MEH data had a more balanced sex distribution, with 52.8% female participants compared to 36.8% in the SDPP data. MEH data was ethnically diverse with individuals identifying as White (45.9%), Asian or Asian British (18.7%, of which 0.6% were Chinese), Black or Black British (9.2%), Mixed (0.9%), other ethnicity (12.7%) and not reported (NR, 12.6%), based on the ethnicity grouping by the UK Office for National Statistics. In contrast, the SDPP cohort comprised only Chinese participants. Imaging devices also varied between the two datasets, with MEH primarily using the 3DOCT-2000SA (Topcon), FD-OCT (Topcon), and CIRRUS (ZEISS), while the SDPP data was collected using TRC-NW300 and TRC-NW400 (Topcon). MEH data, as a hospital-based cohort, exhibits higher disease prevalence than SDPP data, which is derived from community screening. The prevalence of diabetic retinopathy, diabetic macular oedema, and ischaemic stroke in the MEH cohort is 8.71%, 3.02%, and 1.57%, respectively, compared with 1.4%, 0.08%, and 0.66% in the SDPP cohort.

To analyse latent features, we extracted features from 5000 random samples respectively from each site and visualised them using t-distributed stochastic neighbour embedding (t-SNE), a dimensionality reduction algorithm widely used for visualising high-dimensional data[38]. As shown in Fig. 2b and Supplementary Fig. 1, the t-SNE plots revealed clear clustering patterns between the MEH and SDPP cohorts, underscoring the distinct data distributions of the two sites. Furthermore, we measured clinically meaningful morphological indices, such as vascular fractal dimension, which have been proven to be highly associated with systemic conditions like cardiovascular[39–41] and neurological health[42–45]. Figure 2c highlights significant differences in these indices between the two cohorts. For instance, the artery fractal dimension in the MEH dataset is 1.25 (95% CI 1.24, 1.25), compared to 1.28 (95% CI 1.28, 1.29) in the SDPP dataset ($p < 0.001$). More morphological indices are listed in Supplementary Fig. 2.

For FM pre-training, we included two representative and widely used self-supervised learning strategies, generative-based learning (Masked Autoencoder[46]) and self-distillation (DINOV2[47]). We organised downstream tasks using held-out data (i.e. isolated from FM pre-training data) from MEH and SDPP, as well as publicly available datasets sourced from several countries. We adapted FMs to downstream tasks via both fine-tuning (all model parameters tuned on downstream labelled data) and linear probes (all model parameters frozen with one linear classifier tuned on downstream labelled data), as shown in Supplementary Fig. 3. All task performances were assessed using the area under the receiver operating characteristic curve (AUROC) and the area under the precision-recall curve (AUPRC). Details of the downstream task datasets are listed in Supplementary Data 1 and 2. More details about data curation, model adaptation, and model evaluation are introduced in the Methods section.

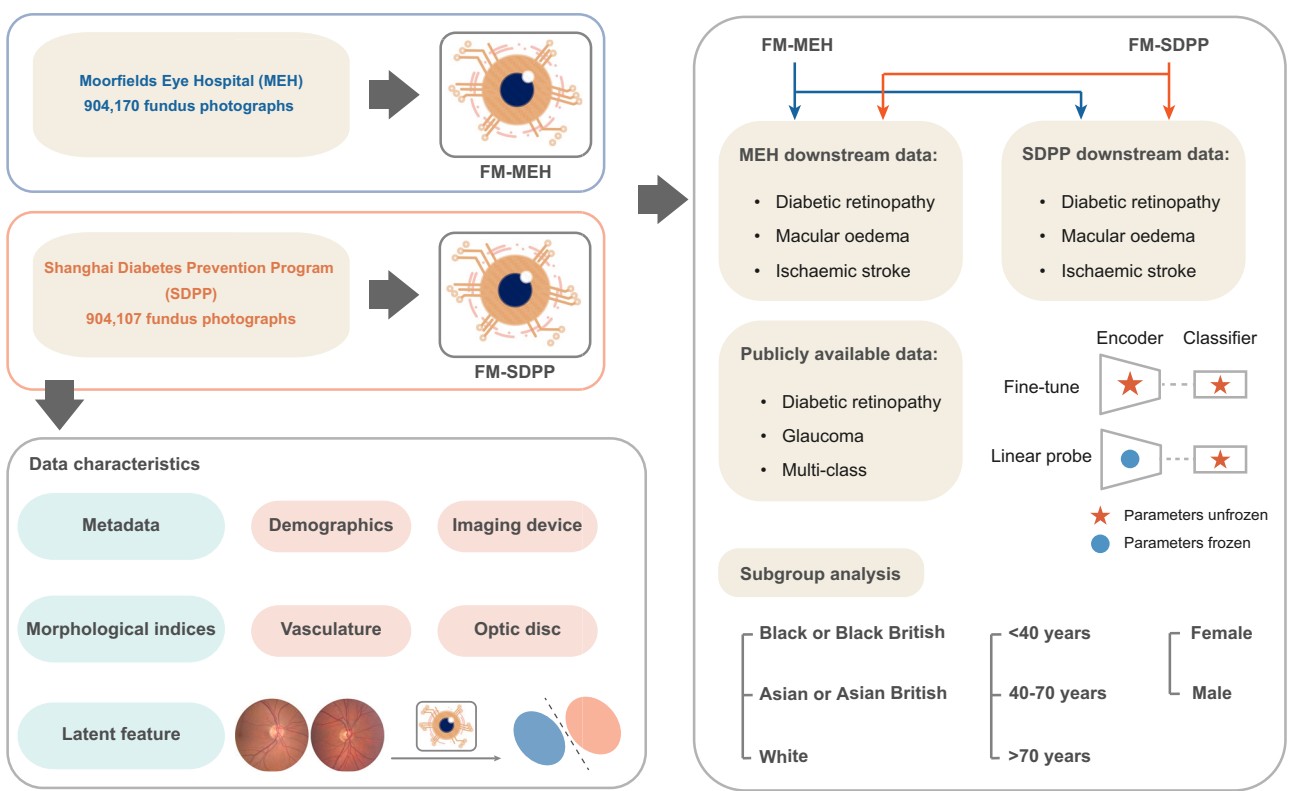

**Fig. 1 | Schematic of the study.** We investigated the impact of pre-training data characteristics on retinal foundation models. Foundation models were pre-trained respectively with data from Moorfields Eye Hospital (FM-MEH) and Shanghai Diabetes Prevention Program (FM-SDPP) and are adapted to downstream tasks for disease detection and prediction. The data characteristics were described in terms of demographic and imaging metadata, clinically meaningful morphological indices, and latent features (representative features encoded by models). Subgroup analysis was performed to evaluate FM fairness over age, sex, and ethnicity.

### Generalisability on downstream tasks curated from each site

We compared FM-MEH and FM-SDPP on three clinically relevant applications (diabetic retinopathy detection, diabetic macular oedema detection, and ischaemic stroke prediction) using held-out MEH and SDPP data. As shown in Fig. 3, the FMs demonstrated good generalisability, with FM-MEH and FM-SDPP showing comparable performance on most evaluations. When adapted to SDPP downstream tasks (Fig. 3a, b), FM-MEH and FM-SDPP performed comparably on all evaluations. Similarly, when adapted to MEH downstream tasks, FM-MEH significantly outperformed FM-SDPP on only three evaluations (ischaemic stroke prediction tasks), as shown in Fig. 3c, d. AUPRC performance followed a similar trend, illustrated in Supplementary Fig. 4. All quantitative results are listed in Supplementary Data 3.

### Generalisability on downstream tasks from publicly available datasets

We evaluated the generalisability of FM-MEH and FM-SDPP to diverse applications using six publicly available datasets, comprising diabetic retinopathy detection (APTOS2019, IDRiD, and MESSIDOR2), glaucoma detection (Glaucoma fundus), and multiple retinal disease detection (JSIEC and Retina). As shown in Fig. 4, FM-SDPP and FM-MEH showed comparable performance in a majority of evaluations (8 out of 12 for Masked Autoencoder-based FMs; 10 out of 12 for DINOV2-based FMs). When pre-trained with Masked Autoencoder and fine-tuned to the downstream tasks (Fig. 4a), FM-SDPP significantly outperformed FM-MEH on three out of six datasets (APTOS2019, MESSIDOR2, and Retina datasets), while the average performance gap on APTOS2019 is minor. When adapted to downstream tasks using linear probing (Fig. 4c), FM-MEH performed significantly better on the Glaucoma fundus dataset ($p < 0.01$). When using DINOV2 for FM pre-training (Fig. 4b, d), FM-MEH significantly outperformed FM-SDPP in two applications, i.e. fine-tuning to MESSIDOR2 ($0.01 < p < 0.05$) and linear probing to IDRiD ($p < 0.01$). AUPRC results are illustrated in Supplementary Fig. 5. All quantitative results are listed in Supplementary Data 3.

### FM fairness over demographic subgroups

We evaluated FM fairness across demographic attribute subgroups in downstream tasks. Using a curated MEH dataset (details introduced in Supplementary Data 4), we examined subgroup performance on diabetic retinopathy detection and diabetic macular oedema detection, considering FM-SDPP and FM-MEH showed no significant differences in overall performance on these tasks (Fig. 3c), and MEH data includes multiple ethnical groups. The SDPP pre-training data is mainly distributed over the young (<40 years) and middle-aged groups (40–70 years), with a subgroup ratio of (0.279, 0.685, 0.036), while MEH data is skewed towards middle-age and older cohorts (>70 years), with a ratio of (0.015, 0.467, 0.518). As shown in Fig. 5a, FM-SDPP performed consistently better than FM-MEH in the young group (grey bars) but worse in the aged group (yellow bars). For instance, in diabetic retinopathy detection using a linear probe, FM-SDPP outperformed FM-MEH by an average AUROC of 0.073 in the young group ($p < 0.01$) while underperforming by 0.047 in the aged group ($p < 0.01$). FM-MEH and FM-SDPP demonstrated similar performance in the middle-aged

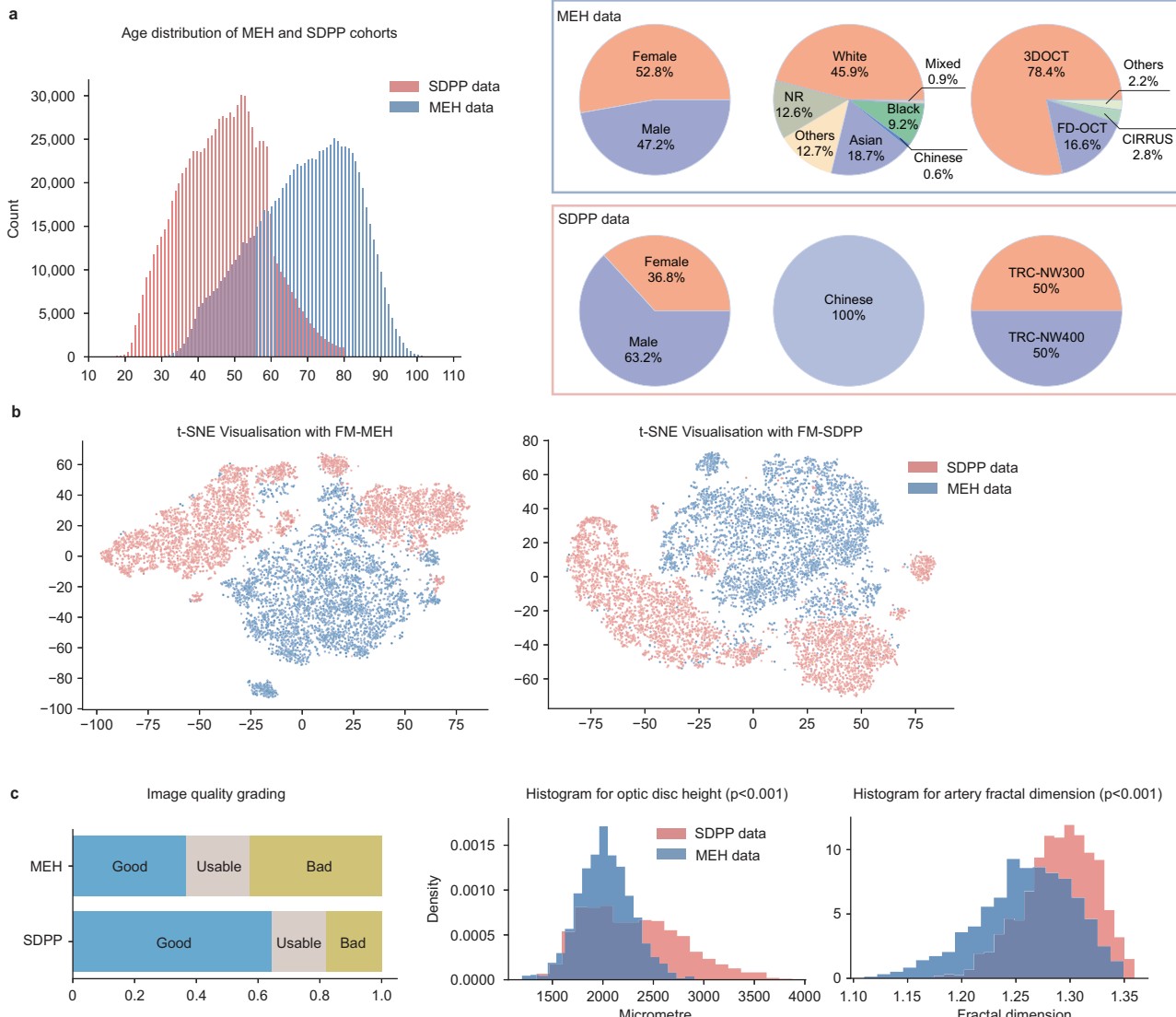

**Fig. 2 | Quantification of data characteristics from Moorfields Eye Hospital (MEH) and Shanghai Diabetes Prevention Program (SDPP).** All values and ratios were calculated over sampled images. **a** The distribution of metadata including age, sex, ethnicity, and imaging devices. **b** t-SNE visualisation for MEH and SDPP data (5000 randomly sampled data points), respectively with features extracted by foundation models developed in each site (FM-MEH and FM-SDPP) with Masked Autoencoder. The distribution of latent features extracted by FMs developed using DINOV2 is shown in Supplementary Fig. 1. **c** The image quality distribution and morphological indices of MEH and SDPP data (5000 randomly sampled data points), obtained with AutoMorph. These demonstrate the distinct distribution of data from Moorfields Eye Hospital and SDPP in multifaceted and complementary views. The *p*-value was calculated by two-sided Welch's *t* test, followed by Holm–Bonferroni correction (*n* = 2). Significant difference in the optic disc height (*p* = 8.07E−102) and artery fractal dimension (*p* = 1.70E−97) are observed. Regarding disease prevalence, MEH (hospital-based) shows higher disease prevalence than SDPP (community-based): diabetic retinopathy 8.71%, diabetic macular oedema 3.02%, and ischaemic stroke 1.57% in MEH vs 1.4%, 0.08%, and 0.66% in SDPP.

group. This consistent performance gap across diverse applications demonstrated the FM bias introduced by differential age distribution in pre-training data.

For ethnicity, despite being pre-trained exclusively on data from the Chinese cohort, FM-SDPP sometimes outperformed FM-MEH in White and Black subgroups on downstream tasks. For instance, FM-SDPP achieved an AUROC of 0.111 higher than FM-MEH in the White subgroup for diabetic macular oedema detection (*p* < 0.01, Fig. 5b). When adapted to diabetic retinopathy detection with the linear probe, FM-MEH pre-trained with DINOV2 significantly outperformed FM-SDPP in Asian cohorts (*p* < 0.01). The performance differences across ethnicity subgroups showed no consistent pattern and did not correlate with the ethnicity distribution of the pre-training data.

For sex, FM-MEH and FM-SDPP showed varying performance across sex subgroups depending on the task and adaptation method (Fig. 5c). Although FM-SDPP pre-training data has a less balanced sex distribution (36.8% female participants versus 52.8% in MEH data), it performed better in female cohorts for certain tasks like diabetic macular oedema detection when pre-trained with DINOV2 and adapted with linear probing. These results showed no correlation with the sex distribution in the pre-training data. All quantitative results and *p*-values are listed in Supplementary Data 5.

**Probing the reasons behind fairness and generalisability**

We probed the reasons why demographic attribute unevenly shaped the model performance, particularly fairness over subgroups. We mainly investigated this from two aspects, the morphological indices

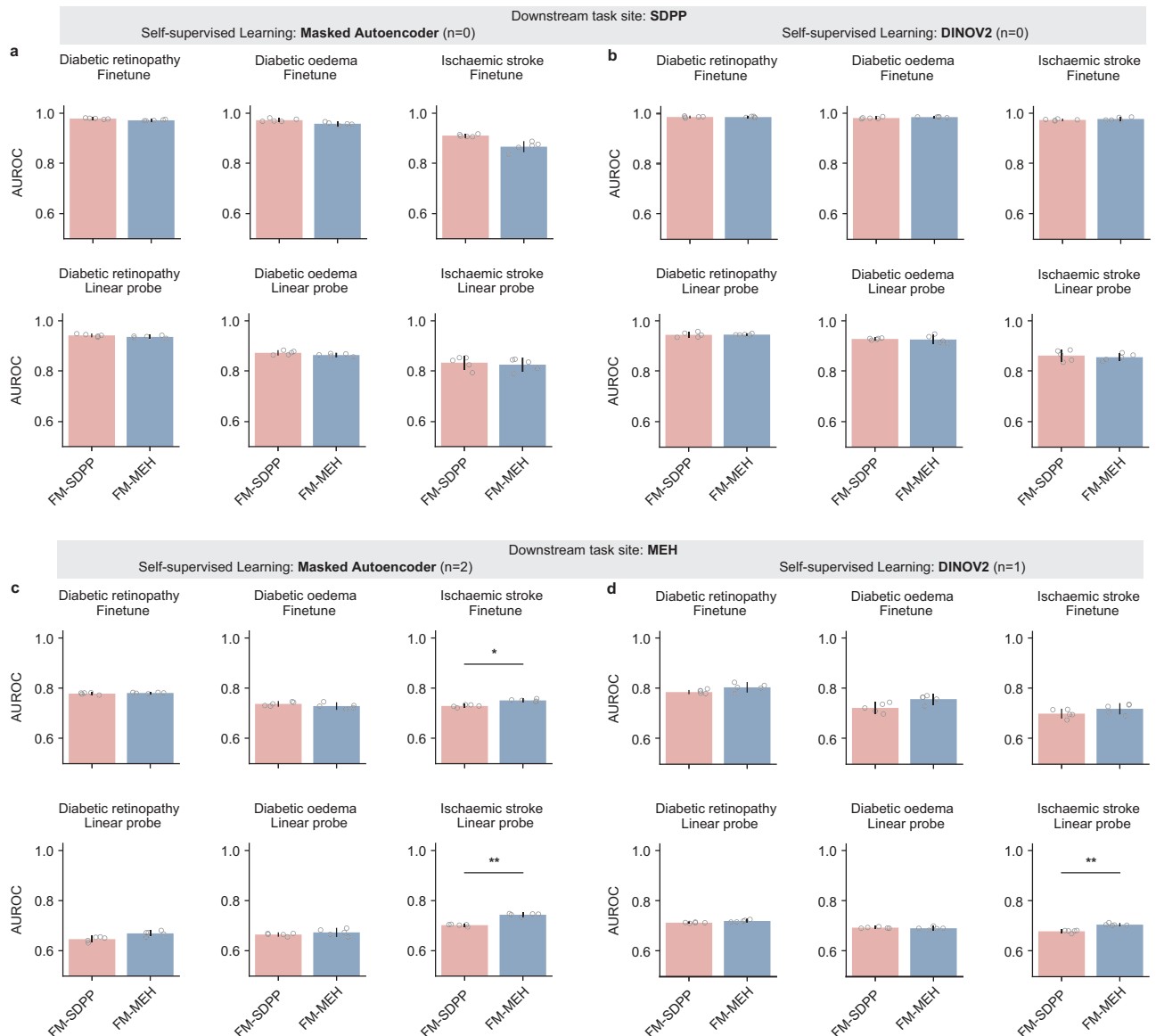

**Fig. 3 | AUROC performance of FM-MEH and FM-SDPP on downstream tasks using data from each site.** Subgraphs **a** and **b** show the model performance on tasks at the SDPP site, with FMs respectively pre-trained with Masked Autoencoder and DINOV2. Subgraphs **c** and **d** present the performance of FMs on tasks at the MEH site. In SDPP downstream tasks, FM-SDPP and FM-MEH showed comparable performance, while FM-MEH achieved superior performance on 3 out of 12 evaluations when adapted to MEH downstream tasks. For each task, models were fine-tuned with five different random seeds, controlling the shuffling of fine-tuning data,

and evaluated on the test set to generate five replicates. The mean AUROC values are represented by bar centres, with error bars indicating 95% confidence intervals (CI). A two-sided Welch's $t$ test followed by Holm–Bonferroni correction ($n = 24$) was used to assess whether the performance differences between FM-SDPP and FM-MEH were statistically significant. * indicates $0.01 < p < 0.05$ and ** indicates $p < 0.01$. $n$ indicates the number of cases showing significant differences. All quantitative results, including raw $p$-value and adjusted $p$-value, are included in Supplementary Data 3.

and latent features across demographic subgroups in pre-training data. As shown in Fig. 6, we observed distinct distribution differences across age subgroups in both latent features (Fig. 6a, c) and morphological indices (Fig. 6e, f). After controlling for confounders by looking at only Asian or Asian British females (Fig. 6b, d), there are still significant gaps in morphological indices (Fig. 6f). This demonstrates that age subgroups exhibit clear morphological variations, and even with self-supervised pre-training, retinal FMs learned distinct latent features across age subgroups, which likely caused bias in FM fairness.

For ethnicity, we investigated three subgroups White, Asian or Asian British, and Black or Black British. The t-SNE visualisations (Supplementary Fig. 6a, FMs trained with Masked Autoencoder)

showed clear clustering partially for the White cohort. When FMs were pre-trained with DINOV2 (Supplementary Fig. 6c), latent features showed no distinct clustering across all ethnicities. Additionally, when controlling for confounders, there were no significant differences across ethnic subgroups in morphological indices (Supplementary Fig. 6f).

For sex subgroups (i.e. female, male), we observed no distinct clustering in t-SNE visualisations, either before or after removing confounding variables (Supplementary Fig. 7). Only the vein fractal dimension showed significant differences across the sex subgroups after removing confounding variables (Supplementary Fig. 7f). These findings suggest that, ethnicity and sex subgroups contributed limited observable variations in morphological indices and less distinct latent

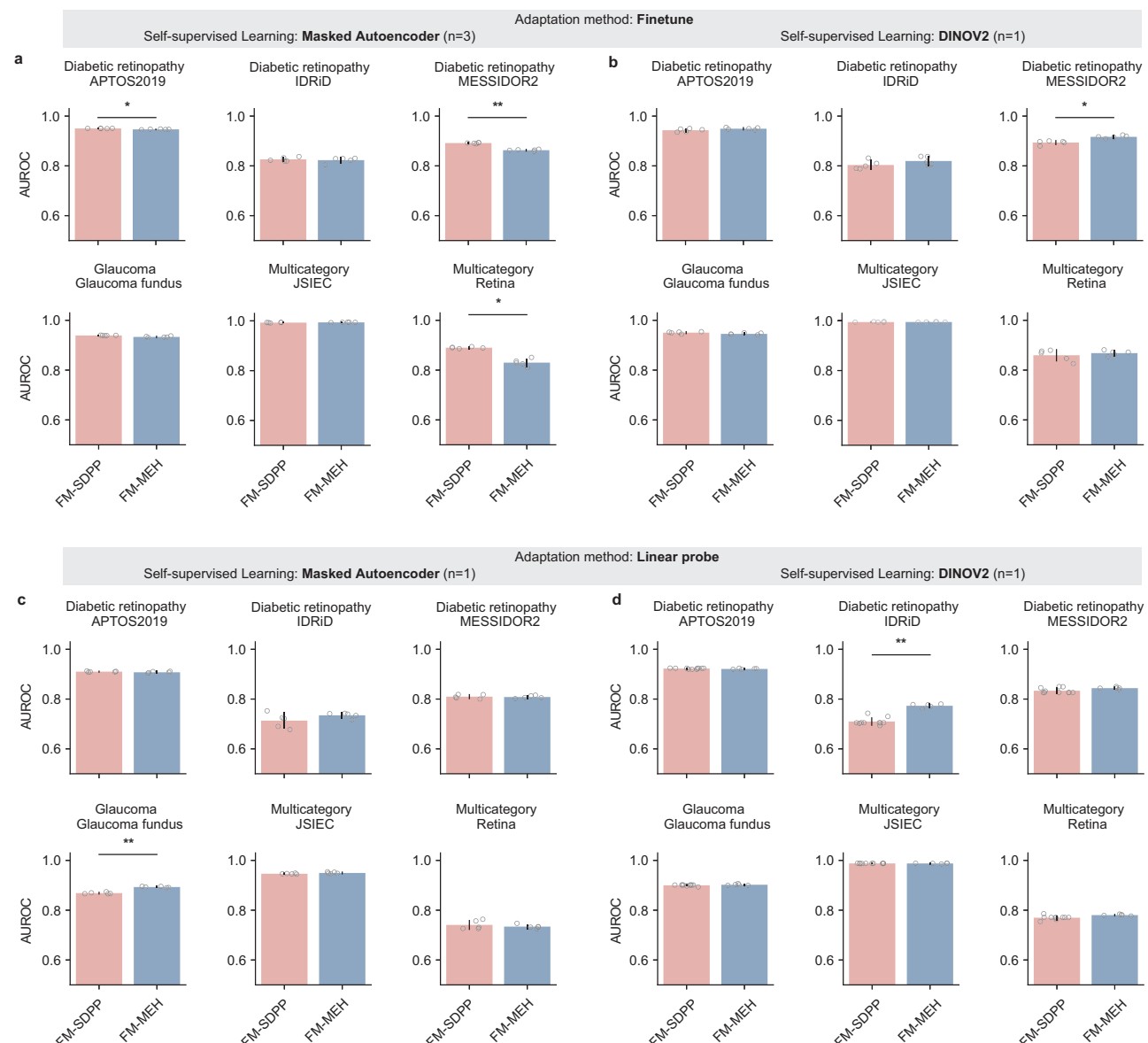

**Fig. 4 | AUROC performance of FM-MEH and FM-SDPP on downstream tasks using publicly available datasets sourced from multiple countries.** Subgraphs **a** and **b** show the performance of FMs pre-trained respectively with Masked Autoencoder and DINOV2 when fine-tuned to downstream tasks. Subgraphs **c** and **d** present the performance of FMs when adapted to downstream tasks with linear probing. When pre-trained with Masked Autoencoder, FM-SDPP significantly outperformed FM-MEH on 3 downstream evaluations. When pre-trained with DINOV2, FM-MEH significantly outperformed FM-SDPP on 2 evaluations. For each task, models were fine-tuned with five different random seeds, controlling the shuffling of fine-tuning data, and evaluated on the test set to generate five replicates. The mean AUROC values are represented by bar centres, with error bars indicating 95% CI. A two-sided Welch's $t$ test followed by Holm–Bonferroni correction ($n = 24$) was used to assess whether the performance differences between FM-SDPP and FM-MEH were statistically significant. * indicates $0.01 < p < 0.05$ and ** indicates $p < 0.01$. $n$ indicates the number of cases showing significant differences. All quantitative results, including raw $p$-value and adjusted $p$-value, are included in Supplementary Data 3.

compared to age distribution, which are less likely to bias FM fairness and generalisability in downstream tasks.

Inspired by the observations above, we tested whether retinal FM generalisability and fairness improve when pre-training on a dataset with a more diverse and balanced age distribution. We generated 300 K synthetic colour fundus photographs representative of SDPP young cohort (age < 40) in age distribution, using synthetic models (details introduced in the Method section) and merged them with randomly sampled 604,170 MEH images, for a total of 904,170 images. Image synthesis was explicitly conditioned on age to match the age profile in SDPP young cohort. The age distribution of this combined data is shown in Supplementary Fig. 8. Using this combined dataset, we

trained FM-MEH-SDPP using DINOV2 and compared it with FM-MEH in subgroup analysis. As shown in Supplementary Data 7, FM-MEH-SDPP performed comparably with FM-MEH, with differences observed for diabetic oedema detection on MEH data and diabetic retinopathy detection on MESSIDOR2 datasets after linear probing. For model fairness across age subgroups (Supplementary Data 8), FM-MEH-SDPP showed significant improvement in young groups for diabetic retinopathy detection after fine-tuning ($p = 0.02$ after Holm–Bonferroni correction) and for diabetic retinopathy after linear probing ($p = 0.033$ after correction). These results indicate that simply increasing age distribution diversity and balance in the pre-training data can yield fairness gains, even with synthetic images.

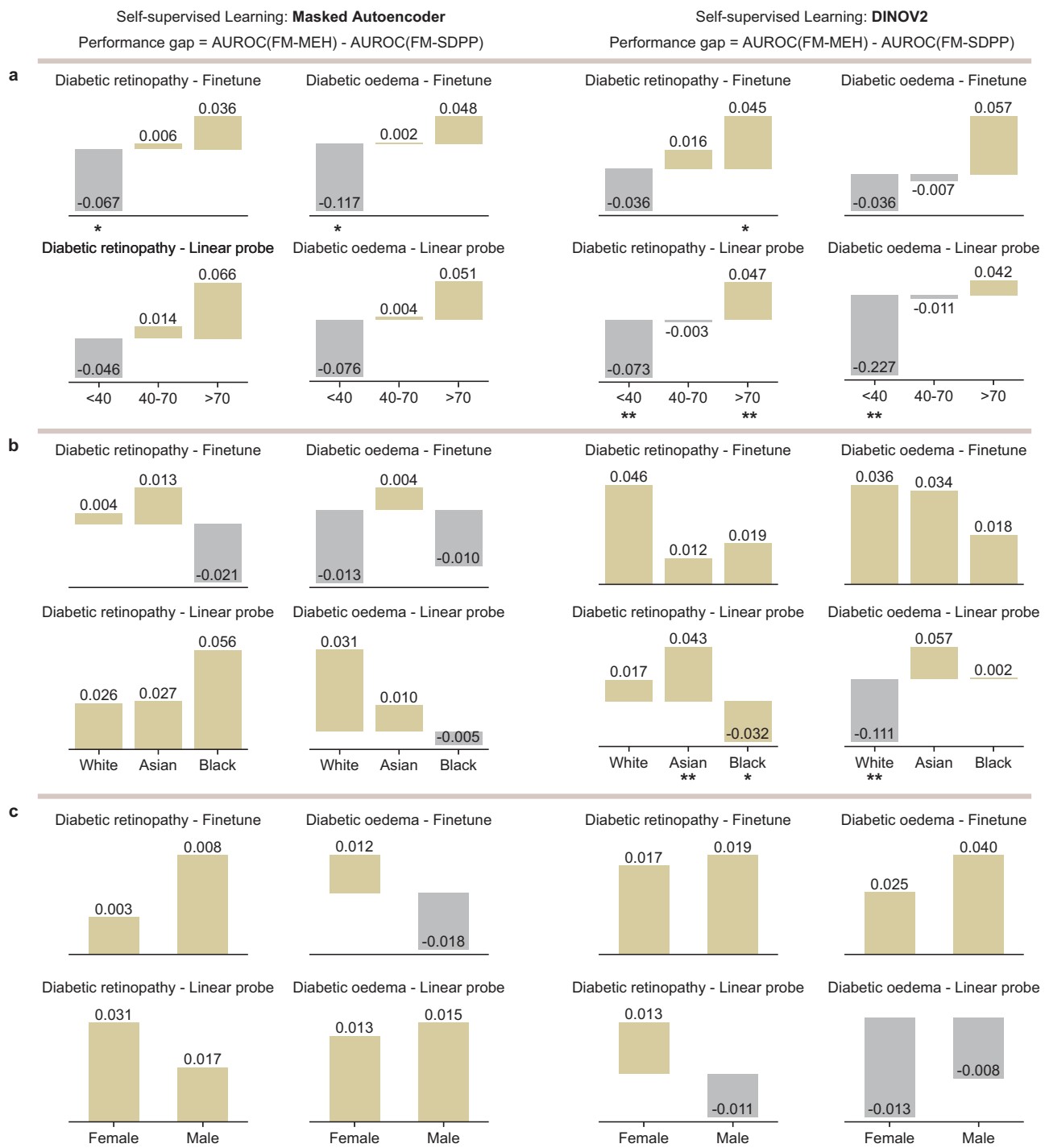

**Fig. 5 | Fairness differences between FM-MEH and FM-SDPP across demographic subgroups.** Performance differences between FM-MEH and FM-SDPP on diabetic retinopathy and diabetic macular oedema detection across age (**a**), ethnicity (**b**), and sex (**c**) subgroups on MEH data. Yellow bars indicate that FM-MEH outperforms FM-SDPP, while grey bars indicate the opposite. The left two columns include the results for FMs pre-trained with Masked Autoencoder, while the right two columns show results for FMs pre-trained with DINOV2. Each section includes the results with fine-tuning and the linear probe. We observed consistent and significant performance gaps between FM-MEH and FM-SDPP in the age section, but not in the sex and ethnicity sections. The bar centres represent the mean AUROC differences. A two-sided Welch's $t$ test followed by Holm–Bonferroni correction ($n = 8$) was used to assess whether the performance differences between FM-SDPP and FM-MEH were statistically significant. * indicates $0.01 < p < 0.05$ and ** indicates $p < 0.01$. All quantitative results, including raw $p$-value and adjusted $p$-value, are included in Supplementary Data 5.

## Discussion

This study investigated the impact of pre-training data on retinal FM performance, particularly generalisability and fairness, a critical yet underexplored area despite the rapid advancements in FM research. We developed parallel retinal FMs with identical implementations that differed only in their pre-training data, so as to isolate the pre-training data effect and provide robust evidence. Our results show that retinal FMs have good generalisability, achieving comparable performance in inter-site adaptation compared to intra-site one. Despite this, fairness gaps were observed across age subgroups, but not across sex or

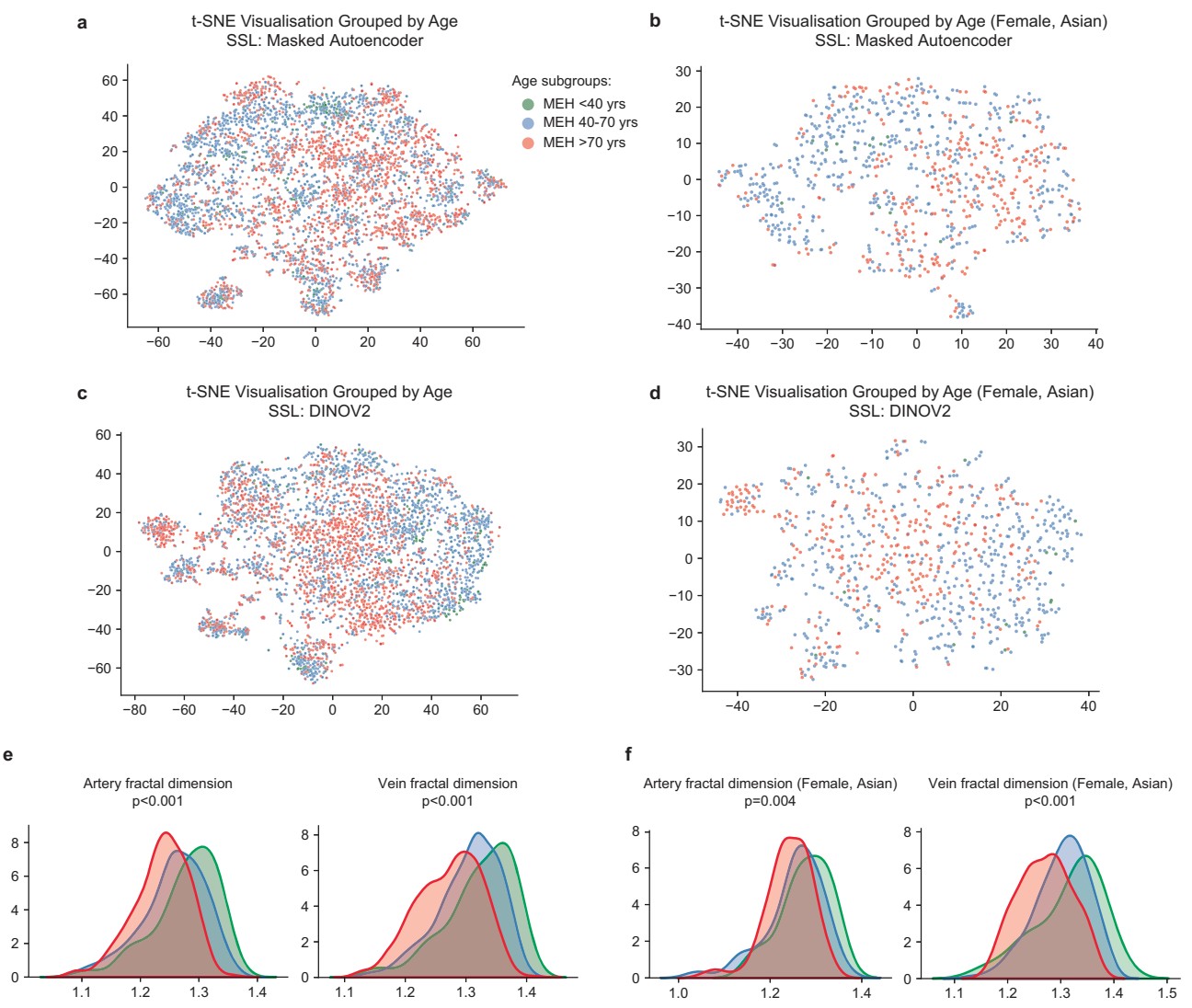

**Fig. 6 | Distribution of latent features and clinically meaningful morphological indices in MEH cohort. a** Shows t-SNE visualisation with latent features across age subgroups extracted by FMs trained with Masked Autoencoder, **b** shows t-SNE visualisation after eliminating confounding effects by specifying sex and ethnicity (e.g. Female, Asian or Asian British). **c, d** Show t-SNE visualisation with latent features extracted by FMs trained with DINOV2. **e** Demonstrates the distribution density of clinically meaningful morphological indices over age subgroups, while

**f** shows the distribution density after specifying sex and ethnicity. A Kruskal–Wallis H-test followed by Holm–Bonferroni correction ($n = 2$) was conducted to assess statistical significance. Both artery fractal dimension ($p = 7.04E{-}5$) and vein fractal dimension ($p = 1.26E{-}5$) show significant differences, even after controlling the sex and ethnicity, artery fractal dimension ($p = 0.004$) and vein fractal dimension ($5.11E{-}4$). These indicate strong data variations across age subgroup.

ethnicity, potentially driven by stronger age-related data variation captured by the FMs. These findings suggest fine-grained, attribute-level data analysis of pre-training data, working as quantitative evidence for efficient data selection in FM development.

Retinal FMs serve as robust base models for diverse applications, as evidenced by the strong generalisability of FM-MEH and FM-SDPP when adapted to downstream tasks across sites. In most evaluations across MEH and SDPP sites, there were no significant differences in performance between intra-site and inter-site adaptations. For example, FM-MEH and FM-SDPP performed comparably on all evaluations on SDPP tasks, and 9 out of 12 on MEH downstream tasks (Fig. 3), showcasing great generalisability given the substantial differences between SDPP and MEH data (Fig. 2). Unlike prior studies that evaluated generalisability by comparing FMs to traditional application-specific models, our approach provided a robust analysis by comparing the generalisation performance between intra-site (e.g. FMs pre-trained on SDPP data and adapted to SDPP downstream tasks) and

inter-site (e.g. FMs pre-trained on MEH data and adapted to SDPP downstream tasks) adaptation. When adapted to publicly available datasets, FM-MEH and FM-SDPP performed comparably in a majority of evaluations (Fig. 4). The generalisation gap narrows further when we disregard the minor differences in certain applications (e.g. fine-tuning to APTOS2019). Such a level of generalisability is rarely observed in traditional application-specific AI models, which typically perform significantly better in internal sites compared to external ones, even after fine-tuning or linear probing. Our study provides robust evidence for the broad generalisability of retinal FMs, reinforcing their potential as base models for diverse adaptations in varied environments.

Retinal FMs are more likely to demonstrate a generalisation gap in clinical tasks that hinge on subtle morphological manifestations, e.g. ischaemic stroke prediction in our study. FM-MEH significantly outperformed FM-Shanghai only for ischaemic stroke prediction, using both masked autoencoder and DINOv2 pre-training strategies (Fig. 3). Compared to ocular disease detection (e.g. diabetic retinopathy and

diabetic oedema), which targets well-defined lesions with abnormal colour and texture patterns, predicting systemic diseases from retinal images (termed as Oculomics[48,49]) typically depends on subtle changes in retinal anatomical tissues, such as vasculature and optic disc, arising from metabolic and circulatory dysfunction. Prior work has shown that retinal vascular calibre, tortuosity, fractal dimension, and optic-disc features carry systemic risk signals[39–45]. For example, narrower retinal arterioles and wider venules have been associated with long-term risk of mortality and ischaemic stroke (both sexes) and coronary heart disease (in women)[41], and patients with Alzheimer's disease exhibit narrower venules with sparser, more tortuous vasculature compared with controls. Our results point to greater generalisation challenges for such Oculomics tasks and, for the time being, support using retinal FMs pre-trained on local data when available for these applications. Additionally, we measured the squared Maximum Mean Discrepancy ($MMD^2$) between each public dataset and the MEH and SDPP cohorts (Supplementary Data 9) and visualised feature distributions using t-SNE (Supplementary Fig. 9). $MMD^2$ offers partial explanatory power for the generalisation gap between FM-MEH and FM-SDPP. For example, with MAE-based features, MESSIDOR2 distributes closer to SDPP than to MEH and FM-SDPP performs better; with DINOv2-based features, MESSIDOR2 is closer to MEH and FM-MEH performs better. However, this pattern does not hold for all datasets (e.g. Retina), indicating that distance metrics alone are insufficient to account for performance and generalisability differences. While FMs substantially mitigate cross-site generalisation gaps, residual disparities warrant further investigation.

Pre-training data attributes, such as demographics, shape retinal FM generalisability and fairness differently, highlighting the need for fine-grained, attribute-level data analysis and selection. Previous studies have explored how labelled data influences the performance of application-specific models. For instance, a recent study[40–44,50,51] showed that fine-tuning data with uniformly sampled demographic attributes (e.g. age, sex, and ethnicity) improved model fairness in clinically relevant applications. However, whether these attributes in pre-training data significantly influence FM fairness remains an open and urgent question, as it defines the foundational capabilities of FMs. In our study, subgroup analysis revealed that the age distribution introduced bias in model fairness across downstream tasks (Fig. 5), while similar biases were not observed for sex or ethnicity. This suggests that, at least for retinal images, simply increasing range and balance of ethnicity and sex may not necessarily enhance FM fairness in downstream tasks, while ensuring a balanced and wide-ranging age distribution can improve both fairness and generalisability. These findings underscore the necessity of fine-grained data analysis for efficient data sampling in medical FM development, before uniformly scaling data across all metadata dimensions.

Data attributes that drive substantial variations in morphological indices and latent features are likely to affect FM generalisability and fairness. Morphological indices quantify the clinically meaningful variations collectively influenced by multiple factors, including demographics, ocular disease phenotypes, and systemic conditions. They provide a clinically interpretable view of data distribution and have been extensively studied in clinical association research[39–43,50,52]. Meanwhile, latent features represent how models perceive the data and are highly relevant to model performance in downstream tasks. Prior machine learning research has characterised data distribution in latent space and regularised features for generative modelling[51,53] and domain adaptation[54,55]. Here, we demonstrate that morphological indices and latent features offer complementary descriptions of pre-training data. In our retinal FM setting, age distribution produced significant subgroup differences in both morphological indices and latent features, and it significantly impacted the FM generalisability and fairness.

Advancements in general-purpose AI methods continue to push the performance boundaries in our applications. In this study, we incorporated two representative self-supervised learning paradigms, a generative-based method (Masked Autoencoder[47]) and a self-distillation-based method (DINOv2[48]), to develop RETFound-MAE and RETFound-DINOv2, using the same compute budget (identical GPU settings and training duration). Our experiments showed that RETFound-DINOv2 achieved superior performance in downstream tasks across both sites (Supplementary Data 6), and this trend extended to publicly available datasets. This advantage likely reflects that strong initial weights from general-purpose AI models, such as DINOv2, can facilitate subsequent domain-specific foundation model development and applications. However, it should be noted that the retinal foundation models developed in this study, RETFound-MAE and RETFound-DINOv2, were not optimised to their theoretical best. Both were pre-trained out-of-the-box on large-scale retinal datasets without extensive hyperparameter tuning. Even so, they achieved competitive performance across diverse downstream applications, highlighting the translational potential of medical foundation models initialised from evolving general AI models.

This study provides quantitative evidence that the composition of pre-training data shapes the generalisability and fairness of retinal FMs. Some of our findings align with prior work on generalisability and fairness in medical imaging. For example, Yang et al. showed that models encoding more demographic attributes (e.g. age, sex) are prone to fairness gap under external evaluation across radiology, dermatology, and ophthalmology[56]. We extend this observation to FMs. As shown in Figs. 5 and 6, differences in age distribution within the pre-training set are perceived by retinal FMs (age < 40, 40 < age < 70, 70 < age) and fairness gap across age subgroups is observed. Kerem et al. verified the good generalisability of vision FMs in medical image segmentation[24]. We also confirmed strong generalisability of retinal FMs. Importantly, rather than comparing FMs to baselines in external evaluation, we trained parallel FMs and conducted head-to-head intra-site vs inter-site comparison, showing that external performance matches internal performance, a stricter demonstration of generalisability. Wang et al. used auxiliary demographic tasks to improve external performance[25], while in our study, we verified that retinal FMs are highly generalisable. To our knowledge, this is the first study to use parallel retinal FMs to rigorously isolate the impact of pre-training data on both generalisability and fairness, an urgent and under-studied question. Together, these results both converge with the existing literature and advance it in FM area by showing how targeted pre-training curation can steer fairness improvement in medical foundation models.

Our experiments with FM-MEH-SDPP echo prior findings that synthetic data can contribute effectively to FM pre-training[57,58]. Moreover, our fine-grained, age-conditioned image synthesis further verified the benefits of increasing the diversity and balance of age, a demographic attribute that strongly impacts both morphological indices and latent features. Although FM-MEH-SDPP performed comparably with FM-MEH on the general cohort, it significantly improved the performance on younger cohorts on certain applications. While synthetic images cannot perfectly recapitulate real distributions, and we conditioned on age only, the observed subgroup improvements indicate that age is indeed an important metadata that should be prioritised in data selection for retinal foundation models. Nevertheless, signals on FM generalisability and how pre-training data impacts FM fairness may differ across medical domains and warrant domain-specific investigation, which we advocate through this study.

Although this work systematically reveals the impact of pre-training data on FM performance using real-world clinical data, several limitations and challenges remain to be addressed in future research. First, although this work describes data distribution in a multifaceted view: demographic and imaging metadata, latent features, and

morphological indices, future studies should include extra factors particularly concerning disease phenotypes. This is currently limited by the complexity of disease categories and severity, as well as challenges in precisely controlling disease phenotypes in large-scale pre-training data organisation. Second, due to the considerable workload involved in developing parallel FMs and organising diverse downstream tasks, this study primarily focused on representative self-supervised learning strategies, such as Masked Autoencoder and DINOV2, and used retinal images as an exemplar. Whether good generalisability exhibited by retinal FMs and uneven impact of data attributes extend to other task types (e.g. segmentation, regression tasks) and medical fields (e.g. pathology, radiology) is unknown. Further research involving a wider range of learning strategies, task types, and medical domains with a replicated parallel design is needed. Third, due to the differences in the sources of labels (e.g. MEH diabetic retinopathy labels were extracted from clinical practice records; SDPP labels were annotated by two ophthalmologists with disagreements adjudicated by a consultant-level ophthalmologist), there are clear performance differences across various applications, as shown in Fig. 3. Although this does not bias the performance comparison between FM-MEH and FM-SDPP, a well-aligned labelling system would enable extra cross-validation. Fourth, we balanced the downstream tasks by sampling equal numbers of disease and control images. This design facilitates fair model comparison and subgroup analyses. However, future evaluations should also include datasets that reflect community prevalence to assess real-world performance. Coupled with analysis at clinically relevant precision–recall operating points, this would provide deeper insight into performance trade-offs. Building upon this study, future work could quantify the extent to which our findings can improve efficiency in FM development, such as proposing practical data sampling methods and quantifying saved resources, including data volume and computation resources.

In conclusion, we unravel the impact of pre-training data on the performance of retinal FMs, demonstrating that both AI fairness and generalisability start at the foundations–the pre-training data. Establishing an accurate and clear understanding of this knowledge is crucial to optimising the development and use of FMs. Our findings suggest that more attention and efforts are required for pre-training data analysis and selection, specifically understanding fine-grained, attribute-level impact for efficient data sampling in medical FM development.

## Methods

### Ethics statement
Ethical approval for this study was obtained from the London-Central Research Ethics Committee (18/LO/1163, approved 1 August 2018), Advanced statistical modelling of multimodal data of genetic and acquired retinal diseases (20/HRA/2158, approved 5 May 2020), Confidential Advisory Group for Section 251 support (18/CAG/0111, approved 13 September 2018), and the Ethics Committee of Shanghai Sixth People's Hospital (Approved No: 2019-087, approved 29 August 2019). The National Health Service Health Research Authority gave final approval on 13 September 2018. Moorfields Eye Hospital NHS Foundation Trust validated the de-identifications for MEH data. Only de-identified retrospective data were used for research.

The requirement for informed consent using SDPP data was waived by the Shanghai Sixth People's Hospital ethics committee because the study used retrospectively collected, anonymised imaging data. The requirement for informed consent using AlzEye data was waived following section 251 supported by the UK CAG because the study used retrospectively collected, anonymised imaging data.

### Source for pre-training data
This research complies with approved ethical regulations. The Moorfields Eye Hospital (MEH) cohort was sourced from AlzEye[59], a retrospective cohort study linking ophthalmic data from 353,157 participants, who attended MEH between 2008 and 2018, with systemic health data from hospital admissions across the whole of England. The ethnicity groups are reported based on the ethnicity grouping by the UK Office for National Statistics. The Shanghai Diabetes Prevention Program (SDPP) cohort was drawn from a community-based longitudinal study of 79,284 participants who underwent physical examinations at Huadong Sanatorium and Shanghai Sixth People's Hospital between December 2015 and November 2022. We randomly sampled 904,170 retinal fundus photographs from each database for FM pre-training. The corresponding data characteristics are listed in Fig. 2. The retinal fundus photographs have a normal field of view (<60°), i.e. no ultra-widefield fundus images were used.

### Data for downstream tasks
We evaluated the foundation model performance on clinically relevant applications using the data from Moorfields Eye Hospital (MEH) UK, SDPP China, and publicly available datasets. First, we organised ocular disease detection tasks, including diabetic retinopathy and diabetic macular oedema detection, using MEH and SDPP data which were held out from FM pre-training data at the patient level. There was no overlap of patients between pre-training and downstream data. We curated 2000 images with labels of diabetic retinopathy and macular oedema from 2000 participants. The labels for diabetic retinopathy are based on the International Clinical Diabetic Retinopathy Severity scale[60], indicating five stages from no diabetic retinopathy to proliferative diabetic retinopathy. The 2000 images were evenly distributed over the five categories. The labels for diabetic macular oedema included three categories: no diabetic oedema, non-clinically significant diabetic macular oedema, and clinically significant diabetic oedema[56,61]. For MEH data, the labels were obtained from clinical practice records. For SDPP data, two independent ophthalmologists annotated the disease labels, with disagreements adjudicated by a consultant-level ophthalmologist. Second, we curated the task of ischaemic stroke prediction using MEH and SDPP data. The stroke labels include binary categories, i.e. no stroke event within three years from imaging or stroke event within three years. For SDPP data, stroke labels were obtained from digital hospital records and self-report records during longitudinal visits between December 2015 and November 2022. For MEH data, systemic health data were derived from Hospital Episode Statistics (HES) data relating to admitted patient care (inpatient records). Diagnostic codes in HES admitted patient care were reported according to the tenth revision of the ICD (International Statistical Classification of Diseases)[62]. ICD codes for stroke (I23-I24) were used in line with previous reports. The stroke data from MEH included 2526 images with each category having 1263 images, while SDPP data included 2000 images with each category including 1000 images. More details are listed in Supplementary Data 2. For subgroup analysis to understand FM fairness, we increased the sample size for diabetic retinopathy detection and diabetic oedema detection to 10,014. The image numbers for each demographic subgroup are listed in Supplementary Data 4.

Similarly to the RETFound study[63], we organised six ocular disease detection tasks with publicly available datasets. For diabetic retinopathy diagnosis, Kaggle APTOS2019 (India), IDRID (India)[64] and MESSIDOR2 (France)[65] were used, with the labels defined by the International Clinical Diabetic Retinopathy Severity scale. For glaucoma, Glaucoma Fundus (South Korea)[66] was included, with three categorical labels, non-glaucoma, early glaucoma (suspected glaucoma) and advanced glaucoma. For datasets with several diseases, JSIEC (China)[67] and Retina were included. JSIEC included 1000 images with 39 categories of common referable fundus diseases and conditions. Retina had labels of normal, glaucoma, cataract and retina disease. The grading protocols for the public datasets were summarised as: IDRiD, two medical experts provided adjudicated consensus

grades; MESSIDOR2, adjudicated by a panel of three retina specialists in accordance with a published protocol; APTOS2019, Kaggle dataset with limited information but possibly a single clinician grader; Glaucoma Fundus, agreement of two specialists based on visual fields and extensive imaging, and JSIEC, labelled by ophthalmologists and confirmed by senior retina specialists. Disagreements were resolved by a panel of five senior retina specialists. Retina, details not available. The details of datasets, such as imaging devices, country and label category, are listed in Supplementary Data 1.

## Data processing for self-supervised learning

We used AutoMorph[68], an automated retinal image analysis tool, to exclude the background and keep the retinal area. All images were resized to 256 × 256 with cubic interpolation. We followed the default data augmentation settings as Masked Autoencoder and DINOV2. On pre-training with Masked Autoencoder, we included random crop (lower bounds 20% of the whole image and upper bounds 100%) and resized the cropped patches to 224 × 224, random horizontal flipping and image normalisation. For DINOV2, the global patch augmentation included random crop (lower bounds 32% of the whole image and upper bounds 100%) and resizing the cropped patches to 224 × 224, random horizontal flipping, colour jittering (brightness 0.4, contrast 0.4, saturation 0.2, and hue 0.1), followed by either Gaussian blur or Gaussian blur and random image solarising (threshold 128, possibility 20%). The local patch augmentation included random crop (lower bounds 5% of the whole image and upper bounds 32%) and resizing the cropped patches to 96 × 96, random horizontal flipping, colour jittering, and random Gaussian blur (possibility 50%). All augmented patches were normalised. We also measured the image quality and morphological indices with AutoMorph.

## Data synthesis

We trained Stable Diffusion XL[63] to generate synthetic retinal images representative of the SDPP cohort. To improve training efficiency on retinal data, we applied Low-Rank Adaptation[69]. For conditional model training, we randomly sampled 200,000 images from the SDPP cohort aged below 40 years, along with their corresponding ages. After training, we generated 300,000 synthetic images that followed the sampled SDPP age distribution. In parallel, we randomly selected 604,170 images from the previously curated set of 904,170 real-world MEH images. The real and synthetic images were then combined to pre-train retinal foundation models, using the same procedures described in sections below. We note that image synthesis may not always strictly follow the age condition. However, as the SDPP data used for Stable Diffusion XL training are drawn from a young cohort, the resulting synthetic images should still be broadly representative of this group.

## Foundation model implementations

For FM pre-training, we selected two representative self-supervised learning strategies, Masked Autoencoder and DINOV2. Both have been widely used across various domains including medical applications, and have demonstrated state-of-the-art performance in disease diagnosis[6,14,16,70]. We used a specific configuration of Masked Autoencoder comprising an encoder and a decoder. The encoder was a large vision Transformer (ViT-large) with 24 Transformer blocks and an embedding vector size of 1024, while the decoder was a small vision Transformer (ViT-small) with eight Transformer blocks and an embedding vector size of 512. The encoder took unmasked patches (with a patch size of 16 × 16) as input and projected them into feature vectors of size 1024. These feature vectors passed through the 24 Transformer blocks, which consisted of multi-headed self-attention and multilayer perceptrons to generate high-level features. The decoder reconstructed the image by inserting masked placeholder patches into the extracted high-level features and then projecting

them back to image patches through a linear projection layer. During model pre-training, the objective was to reconstruct retinal images from the highly masked version, with a mask ratio of 0.75. The pre-training batch size was 1792 (4 GPUs × 448 per GPU). The total pre-training epoch was 800 and the first 15 epochs were for learning rate warming up (from 0 to a learning rate of $1 \times 10^{-3}$). The model weights at the final epoch were saved as the checkpoint for adapting to downstream tasks.

We specified DINOV2 with both teacher and student networks as ViT-large, with 24 Transformer blocks and an embedding vector size of 1024. It included a projection head of three-layer perceptrons, respectively with dimensions 2048, 384, and 131,072. The patch size was 14 × 14. The teacher network processed the global patches while the student network processed both global and local patches. During model pre-training, the objectives combined the original objectives of DINO[71] and iBOT[72]. The DINO part calculated the cross-entropy loss between the categorical tokens from the teacher network and the student network, while the iBOT part calculated the cross-entropy between masked patch tokens between the two networks (the maximum number of masking patches was 128). The pre-training batch size was 320 (4 GPUs × 80 per GPU). The total pre-training epoch was 100 and the first 10 epochs were for the learning rate warming up (from $1 \times 10^{-6}$ to a learning rate of $2 \times 10^{-4}$) and the remaining 90 epochs for a cosine annealing schedule. The model weights at the final epoch were saved as the checkpoint for adapting to downstream tasks.

## Adaptation to downstream tasks

When adapting foundation models pre-trained with Masked Autoencoder to downstream tasks, we only need the encoder (ViT-large) of the foundation model and discard the decoder. For foundation models pre-trained with DINOV2, the teacher network was used and adapted to downstream tasks. Both the encoder and teacher networks extracted high-level features from retinal images. A fully connected layer took these features as input and output the probability distribution over the disease categories. The category with the highest probability was selected as the final classification. The number of categories determined the number of neurons in the fully connected layer. We used two adaptation strategies, fine-tuning and the linear probe. Fine-tuning tuned the encoder and fully connected layer using the downstream data while the linear probe tuned only the fully connected layer. The schematic diagram is shown in Supplementary Fig. 3. The training objective was to predict the same categorical output as the label. The batch size was set to 16, and the model was trained for 50 epochs. The first 10 epochs followed a learning rate warm-up schedule, increasing linearly from 0 to a learning rate of $5 \times 10^{-4}$. This was followed by a cosine annealing schedule, where the learning rate gradually decreased from $5 \times 10^{-4}$ to $1 \times 10^{-6}$ over the remaining 40 epochs. After each training epoch, the model performance was evaluated on the validation set. The model checkpoint with the highest AUROC on the validation set was saved for subsequent internal and external evaluations.

## Computational resources

Four NVIDIA Tesla A100 (80 GB) were used for self-supervised pre-training in this project. It took about 16 days to finish pre-training with DINOV2 or Masked Autoencoder. We used an equal computational cost from MEH and SDPP for foundation model development. For fine-tuning and linear probing foundation models to downstream tasks, we use NVIDIA Tesla T4 (16 GB). Fine-tuning took about 70 min for every 1000 images, while linear probing took around 15 mins.

## Evaluation and statistical analysis

All task performance was assessed using the classification metrics AUROC and AUPRC. For ischaemic stroke prediction tasks, the

AUROC and AUPRC were calculated in a binary setting. For multiclass classification, such as five-stage diabetic retinopathy and multi-category disease diagnosis, AUROC and AUPRC were calculated separately for each class and then averaged to obtain the overall AUROC and AUPRC scores. For each task, we fine-tuned the model with five different random seeds, which determined the implementations including shuffling of fine-tuning data and data augmentation. The mean and standard deviation of the performance across the five runs were computed. The standard error is estimated as (standard deviation/$\sqrt{5}$), and the 95% confidence interval (CI) is obtained by multiplying the standard error by 1.96. The normality of the model performance was checked via Shapiro–Wilk test. Statistical significance is calculated using two-sided Welch's $t$ test, followed by Holm–Bonferroni correction. When comparing FM-MEH and FM-SDPP, the $p$-value was adjusted to the size of 24 (3*SDPP tasks, 3*MEH tasks, 6*publicly available datasets, each including fine-tuning and linear probing). In subgroup analysis, for each demographic attribute, the $p$-value was adjusted to the size of 8 (2*MEH tasks, 2*Self-supervised learning methods, each including fine-tuning and linear probing). All raw $p$-values, adjusted $p$-values, and significances are listed in Supplementary Data 3–8.

### Reporting summary

Further information on research design is available in the Nature Portfolio Reporting Summary linked to this article.

## Data availability

The MEH data consists of routinely collected healthcare data. Owing to their sensitive nature, the dataset is subject to controlled access by means of a structured application process. The AlzEye dataset is subject to the contractual restrictions of the data sharing agreements between National Health Service Digital, Moorfields Eye Hospital and University College London due to data privacy and ethical requirement. Interested collaborators should contact the chief investigator P.A.K at p.keane@ucl.ac.uk and submit a formal submission at https://www.insight.hdrhub.org/insight-data. The data management committee will then review all the requests and grants. A formal data transfer agreement will be required upon approval. Generally, all these requests for access to the data will be responded to within 1 month. For SDPP data, due to data privacy and ethical requirement, individual-level patient data can be accessible with the consent of the data management committee from institutions and are not publicly available. Requests for the non-profit use of the fundus images and related clinical information should be sent to T.Y.W at wongtienyin@tsinghua.edu.cn. The data management committee will then review all the requests and grants. A formal data transfer agreement will be required upon approval. The requests for access to the data will be responded to within 1 month. Data for ocular disease experiments are publicly available online and can be accessed through the following links: IDRID, MESSIDOR2, APTOS2019, Glaucoma Fundus (https://dataverse.harvard.edu/dataset.xhtml?persistentId=doi:10.7910/DVN/1YRRAC), JSIEC (https://zenodo.org/record/3477553), and Retina (https://www.kaggle.com/datasets/jr2ngb/cataractdataset).

## Code availability

The code used to train, fine-tune and evaluate RETFound from Y.Z. is available at https://github.com/rmaphoh/RETFound, which is based on PyTorch. All pre-trained model weights are available at https://huggingface.co/YukunZhou. Images were processed with automated retinal image analysis tool AutoMorph v.1.0 (https://github.com/rmaphoh/AutoMorph). Stable Diffusion XL can be found at https://huggingface.co/docs/diffusers/en/using-diffusers/sdxl. Results were further analysed and visualised with Python v.3.11.0, NumPy v.1.26.4, SciPy v.1.15.2, Matplotlib v.3.8.4, pandas v.1.5.0, Scikit-Learn v.1.4.2 and Pillow v.10.2.0.

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

## Acknowledgements

Y.Z. is supported by Wellcome Trust Early-career Award (318987/Z/24/Z) and Moorfields Eye Charity Equipment Grant (EQR-25B-102). Y.C.T. is supported by the National Medical Research Council of Singapore (NMRC/MOH/ HCSAINV21nov-0001). C.Y.C. is supported by the Research Grants Council Hong Kong (General Research Fund, ref.14101324). D.C.A. is supported by Engineering and Physical Sciences Research Council (EP/M020533/1, EP/R014019/1 and EP/V034537/1). T.Y.W. is supported by National Natural Science Fund of China (Grant No: 82388101) and Beijing Municipal Science and Technology Program (Grant No: Z241100009024040). B.S. is supported by National Natural Science Foundation of China (Grant No. T2525004). P.A.K. is supported by a UK Research & Innovation Future Leaders Fellowship (MR/T019050/1), the Moorfields Eye Charity with The Rubin Foundation Charitable Trust (GR001753), and an Alcon Research Institute Senior Investigator Award. We acknowledge the computational resources supported by the UCL Computer Science Cluster and UCL Advanced Research Computing UAI platform.

## Author contributions

Y.Z., T.Y.W., B.S., and P.A.K. contributed to the conception and design of the work. Y.Z., Z.W., and Y.W. contributed to the data acquisition and organisation. Y.Z. and Z.W. contributed to the technical implementation. A.Y.O., S.K.W., M.A.C., Z.G., P.N., X.L., Y.X.W., D.A.M., and P.A.K. provided the clinical inputs to the research. Y.Z., Z.W., Y.W., E.R., L.J., J.E., T.L., J.S., K.Z., J.H.L.G., Q.H., Y.C.T., D.C.A., E.J.T., A.K.D., A.A., T.Y.W., B.S. and P.A.K. contributed to the evaluation pipeline and analysis design of this work. All authors contributed to the drafting and revising of the manuscript.

## Competing interests

P.A.K. is a cofounder of Cascader Ltd. and has acted as a consultant for Retina Consultants of America, Roche, Boehringer-Ingelheim, and Bitfount and is an equity owner in Big Picture Medical. He has received speaker fees from Zeiss, Thea, Apellis, and Roche. He has received travel support from Bayer and Roche. He has attended advisory boards for Topcon, Bayer, Boehringer-Ingelheim, and Roche.

## Additional information

[1]Institute of Ophthalmology, University College London, London, UK. [2]NIHR Biomedical Research Centre at Moorfields Eye Hospital NHS Foundation Trust, London, UK. [3]Hawkes Institute, University College London, London, UK. [4]Department of Computer Science, University College London, London, UK. [5]School of Computer Science, Shanghai Jiao Tong University, Shanghai, China. [6]MOE Key Laboratory of AI, AI Institute, Shanghai Jiao Tong University, Shanghai, China. [7]School of Clinical Medicine, Tsinghua Medicine, Tsinghua University, Beijing, China. [8]The Royal Victorian Eye and Ear Hospital, Melbourne, VIC, Australia. [9]State Key Laboratory of Metabolic Dysregulation & Prevention and Treatment of Oesophageal Cancer, Shanghai Key Laboratory of Diabetes Mellitus, Department of Endocrinology and Metabolism, Shanghai Diabetes Institute, Shanghai Clinical Centre for Diabetes, Shanghai Sixth People's Hospital Affiliated to Shanghai Jiao Tong University School of Medicine, Shanghai Jiao Tong University, Shanghai, China. [10]Centre for Innovation and Precision Eye Health; and Department of Ophthalmology, Yong Loo Lin School of Medicine, National University of Singapore, Singapore, Singapore. [11]Ophthalmology and Visual Science Academic Clinical Program, Duke-NUS Medical School, Singapore, Singapore. [12]Singapore Eye Research Institute, Singapore National Eye Centre, Singapore, Singapore. [13]College of Medicine and Health, University of Birmingham, Birmingham, UK. [14]NIHR Birmingham Biomedical Research Centre, Birmingham, UK. [15]Beijing Visual Science and Translational Eye Research Institute (BERI), Beijing Tsinghua Changgung Hospital, Beijing, China. [16]Department of Medical Physics and Biomedical Engineering, University College London, London, UK. [17]Department of Ophthalmology and Visual Sciences, The Chinese University of Hong Kong, Hong Kong Special Administrative Region, Hong Kong, China. [18]Department of Molecular Medicine, Scripps Research, La Jolla, CA, USA. [19]School of Biomedical Engineering, Tsinghua Medicine, Tsinghua University, Beijing, China. [20]These authors contributed equally: Yukun Zhou, Zheyuan Wang, Yilan Wu. ✉e-mail: yukun.zhou.19@ucl.ac.uk; shengbin@sjtu.edu.cn; p.keane@ucl.ac.uk

