## [Peer Review file · Nature Communications]

Understanding Pre-training Data Effects in Retinal Foundation Models using Two Large Fundus Cohorts

Corresponding Author: Dr Yukun Zhou

Version 0:

Reviewer comments:

Reviewer #1

(Remarks to the Author)
Paper Summary

This study investigates the influence of pre-training data characteristics (specifically, geographic origin and associated demographic distributions) on the generalizability and fairness of medical foundation models (FMs) in ophthalmology. The authors pre-train Vision Transformer-based FMs using two self-supervised learning methods (Masked Autoencoder (MAE), and DINOv2) on two large, distinct datasets of retinal fundus photographs: one from Moorfields Eye Hospital (MEH) in the UK and one from the Shanghai Diabetes Prevention Program (SDPP) in China. They evaluate these parallel-trained models (FM-MEH and FM-SDPP) on downstream tasks (diabetic retinopathy, diabetic macular edema, stroke prediction) using held-out data from both sites and several public datasets. Fairness is assessed by comparing performance across subgroups defined by age, sex, and ethnicity using MEH test data.

Main Claims and Observations

Fairness: The paper concludes that for these retinal FMs, the age distribution within the pre-training data introduces measurable performance bias across downstream task age subgroups, while pre-training data distributions for sex and ethnicity did not result in similar consistent biases. [lines 96, 269 following]

Generalizability & Site Bias: The study reports that FMs demonstrate good generalizability, performing comparably across sites and datasets in approximately 70% of evaluations. However, in the remaining ~30% of cases, FMs perform significantly better when the downstream task data aligns with their specific pre-training data source (intra-site advantage). [lines 322 following]

Review Summary

This paper explores how pre-training data influences the behaviour of medical foundation models (FMs), employing a robust parallel training strategy on large-scale, high-quality datasets. While it addresses important questions, its main conclusions add little novelty or practical significance. The finding that models often perform better on data resembling their training distribution (site bias) is already a well-documented principle in machine learning, and the approximately 30% figure reported here lacks deeper explanations of how or why this bias arises.

Furthermore, the finding that age distribution affects fairness, while potentially valid for this specific retinal imaging context, is presented with claims that overreach its generalizability to "medical FMs" broadly. The finding regarding site-specific performance benefits is somewhat inconclusive, lacking a clear framework for practical data selection beyond reaffirming that data distribution matters. Statistical methods require further clarification (especially given multiple comparisons), potentially undermining the reliability of the reported quantitative findings. The authors would benefit from more thorough contextualization of their results relative to previous works on fairness and bias in medical foundation models.

Combined with numerous minor weaknesses, the paper overstates its contributions and requires substantial revision, potentially bordering on rejection due to the limited novelty and potential statistical issues.

Detailed Review

Major Weaknesses

Limited Novelty and Depth of "Site-Bias" Finding

The core finding that models can exhibit better performance on data distributions matching their training data is not novel; it's a fundamental concept in domain adaptation and generalization research. While quantifying this effect for these specific FMs (~30% intra-site advantage) provides a data point, the paper falls short of providing significant new conceptual understanding or predictive insight into when or why this site bias occurs for FMs. This finding also stands in mild tension with literature suggesting large-scale self-supervised training can mitigate certain biases.

Concerns Regarding Statistical Rigor

Multiple Comparisons: The paper reports a large number of statistical tests across many tasks, datasets, models, and demographic subgroups (Figures 3, 4, 6, Supplementary Tables), yet there is no mention of adjusting for multiple comparisons (e.g., via Bonferroni correction). Reporting multiple unadjusted p-values raises the risk of overstating the significance of any observed differences, casting doubt on the reliability of the ~30% site-bias estimate and subgroup findings in Figure 6.

Subgroup Analysis & Sample Size: The authors draw fairness-related conclusions from subgroup analyses (Figure 6). Given the total dataset sizes and the age distribution data cited (Lines 281–282), sample counts within certain subgroups (especially ages <40 and >70) could be critically small, which may render results unstable or prone to outliers. The distribution of test samples and disease prevalence within these subgroups should be explicitly reported.

Prevalence: The potential influence of varying disease prevalence across demographic subgroups (e.g., DR/DME prevalence increasing with age) on subgroup performance metrics (like AUROC) needs clarification and discussion, even if the overall test sets were balanced. Especially in combination with small subgroup sizes <30 (age below 40 in MHE) low prevalence in this group may even result in missing samples for a class.

Overgeneralization of Findings to "Medical Foundation Models"

Scope of "Medical FMs": The paper repeatedly frames its findings in the context of "medical FMs" generally (Title, Abstract, Discussion). However, the empirical results are derived solely from 2D colour fundus photography, specific ViT architectures, and two SSL methods. The conclusion that age is the dominant fairness driver while sex/ethnicity are minimal is specific to this retinal context and cannot be readily generalized. The conclusion that age is the primary driver of fairness while sex and ethnicity show minimal effects applies to this retinal imaging context and cannot be generalized to all medical foundation models. Sex and ethnicity are known to be significant contributors to bias in other medical AI domains (e.g., radiology, dermatology), so any general statements require caution.

Limited Technical Scope: The results may not directly extend to different imaging modalities (CT, MRI), task types (segmentation, image generation), model architectures (CNNs, multimodal models), or other SSL/training methods.

Claims of Offering a "Data Selection Framework":

The paper suggests its findings offer "practical guidance for improving the construction and application of medical FMs" (Lines 319-321) and proposes a "pipeline for key metadata identification" (Fig 5a, Lines 447-448). However, the practical advice boils down to selecting FMs pre-trained on similar data (Lines 339-340), which lacks operational specificity. The flowchart in Figure 5a adds little value; it's a linear description, not a decision-making tool. This falls short of providing a robust framework

Potentially Confounded SSL Comparisons

Comparing MAE and DINOv2 performance is complicated by vastly different reported pre-training protocols (epochs: 800 vs 100; batch size: 1792 vs 320) without discussion of hyperparameter tuning or ablation studies. Observed differences might stem from training budget or suboptimal parameters rather than solely the inherent properties of the SSL methods themselves for this task

Minor Weaknesses

Ambiguity in Figure 6 - The figure does not specify if the test data is from MEH, SDPP, or both combined, nor how the subgroups are distributed within those sets. Details on fine-tuning sets, class distributions, and data balancing are also unclear.

Label Sources and Potential Biases - MEH retinal labels come from clinical records; SDPP labels come from expert annotations. This difference in label acquisition may partly underlie the performance differences, not just the site distribution. The paper briefly mentions it but does not delve into its implications.

Unclear Link to Morphological Features - Although morphological feature differences between MEH and SDPP images are reported, the quantitative relationship between these differences and downstream task performance remains vague. The choice of morphological indices (e.g., fractal dimension) and their direct relevance to diabetic retinopathy, DME, or stroke prediction should be explored.

Missing Pre-training Disease Distribution Details - The paper reports disease prevalence in the downstream sets but not in the pre-training sets. Figure 2 lacks information on the distribution of relevant diseases (DR, DME, stroke indicators if possible) within the MEH and SDPP pre-training datasets, which is highly relevant context. This information on the distribution of relevant diseases (DR, DME, and possibly stroke indicators) in these datasets should be included in Figure 2.

Small Performance Gaps Treated as Significant - Some statistically significant differences reported involve very small absolute AUC changes (e.g., <0.01), questioning their practical relevance and the conclusion of one model outperforming the other. For APTOS2019 AUC difference is 0.003; for Glaucoma fundus 0.006; for Diabetic retinopathy shanghai fine tune 0.006

Contradictory Results Across Architectures - In some cases, FM-SDPP outperforms FM-MEH for one SSL method (MAE) but performs worse under the other method (DINOv2) (e.g., MESSIDOR2, Retina datasets in Fig 4), making it hard to attribute performance solely to dataset differences versus interactions with the pre-training approach. Summing p-values across models is not appropriate.

Figures and Flowcharts -The t-SNE clusterings mentioned in the text (Fig. 5b) are not clearly visible or explained. The flowchart in Figure 5a adds no value; it's a linear description, not a decision-making tool. Lack of discussion on metadata (age, sex, device, etc.) for the public datasets hinders interpretation of performance differences on these sets in the context of pre-training data similarity.

Miscellaneous Issues

Precision Scores: AUPRC is calculated, but precision-recall trade-offs are not discussed in the text.

Figure 2b Specificity: Mentions t-SNE for MAE features but not explicitly for DINOv2 features in the caption/text for comparison.

Typo: Extended Data Figure 4 caption (line 900) likely means AUPRC, not AUROC, based on the figure content.

Image Quality: Plotted in Fig 2c but its impact or relevance is not discussed.

Study Origins: The different clinical origins of the datasets (MEH/AlzEye vs. SDPP/Diabetes Prevention) might imply systematic differences beyond demographics (e.g., comorbidities, baseline health status) that are not explored.

Age Binning: Justification for binning age rather than analyzing it as a continuous variable or using more granular bins is missing.

Strengths:

Parallel Training Design: The core strength is training identical FM architectures with identical methods (per SSL type) on two distinct, large-scale datasets, providing a clean setup to isolate the impact of pre-training data origin.

Valuable Datasets: Access to and use of large, real-world clinical datasets (MEH, SDPP) adds significant value.

Important Problem: The study addresses a timely and critical question about the foundations of generalizability and fairness in medical FMs.

Suggestions

Deeper Bias Analysis: Investigate potential mechanisms behind observed biases. How do models encode ethnicity/gender, even if downstream performance differences weren't consistently significant in this study?

Combined Dataset Training: Explore if pre-training on a combined MEH+SDPP dataset mitigates the observed age bias and site-specific performance gaps.

SSL Comparison Rigor: Provide justification for the chosen training protocols or conduct ablation studies to better isolate the effect of the SSL strategy itself versus training budget/hyperparameters, especially for claims of DINOv2 superiority.

Expand Scope (Future Work): Acknowledge limitations more strongly and suggest future work replicating this parallel design in other medical domains (e.g., radiology, pathology) to assess the generalizability of these specific findings.

Statistical Correction: Re-evaluate statistical significance using appropriate corrections for multiple comparisons and report corrected p-values or use confidence intervals more centrally. Clearly report subgroup sample sizes and prevalence.

Contextualize Novelty: Reframe the "site bias" finding by clearly acknowledging it confirms existing principles and focus novelty on the quantification within the FM/retinal context, or on deeper analysis if performed.

Relevant references:

Practical selection of pre-training data and sub-group analysis to address bias

Yang et al. (Nature Medicine): The limits of fair medical imaging AI in real-world generalization.

Foundation models do generalize

Cekmeceli et al.: Do Vision Foundation Models Enhance Domain Generalization in Medical Image Segmentation?

<https://arxiv.org/pdf/2409.07960>

Mitigating intra- and intersite heterogeneity

Chaudhari et al.: Embracing the Disharmony in Heterogeneous Medical Data.

Bias in non-medical foundation models under long-tailed data
Chen et al.: Rethinking the Bias of Foundation Model under Long-tailed Distribution.
<https://arxiv.org/pdf/2501.15955>

Fairness in DINOv2
Gustafson et al.: FACET: Fairness in Computer Vision Evaluation Benchmark.

Dataset bias in contrastive learning
Purushwalkam et al. (NeurIPS 2020):
https://proceedings.neurips.cc/paper_files/paper/2020/file/22f791da07b0d8a2504c2537c560001c-Paper.pdf

(Remarks on code availability)

Reviewer #2

(Remarks to the Author)

Noteworthy results

Yukun Zhou and co-authors, in "Revealing the Impact of Pre-training Data on Medical Foundation Models," explore fairness and generalizability of medical foundation models, using ophthalmic fundus photography as the prototypical case. The most important finding is the impact of certain metadata on foundation model (FM) generalizability, in particular with regard to age. Since the authors created two different foundation models, one with MEH data and another with SDPP data. The groups differed across a few domains, most notably age. The investigators demonstrated differences between foundation model performance when comparing MEH and SDPP analysis of datasets and also showed the effect of age within models. This is a key finding, as it speaks to the importance of transparency and bias in model fairness and generalizability with regard to the datasets used to create the foundation models.

Ethnicity and sex did not appear to affect model performance. The investigators showed "that FMs demonstrate[d] good generalizability, achieving comparable performance in over 70% of downstream tasks. However, FMs sometimes perform[ed] better on application data that aligns with their pre-training data."

The authors point out that "long-term and sustainable advancement of medical AI requires the development of domain-specific techniques tailored to the unique characteristics of medical data and application scenarios... Pre-training clinical data and self-supervised learning strategies have a synergistic effect on FM performance... need to simultaneously optimise both model learning strategies and pre-training data characteristics to advance medical FM development."

The authors make several excellent and important recommendations for future work in this area, including the following:

- "Future studies should include extra factors particularly concerning disease phenotypes."
- "Further research involving a wider range of learning strategies and medical domains is needed."
- "There are clear performance differences across 435 various applications."

Of note is that the MEH and SDPP samples used to create and test the FMs were unbalanced and a subset of data were used to create balanced and curated samples. The authors should comment on how this might have affected the FMs and comparisons.

Significance to the field

This work has significant potential to enhance the quality of FMs used for analysis of medical data. It is an interesting examination of the role and effect of training data on FM performance and bias.

Conclusions and Claims

The authors' conclusions and claims are supported by their data.

Flaws in the data analysis, interpretation and conclusions

No significant flaws are noted.

Methodology

The methodology is sound, well thought out and well executed. As noted above, the authors should comment on the imbalance of the MEH and SDPP datasets.

Reproduction

The authors have provided adequate information to enable reproduction of their results, although their data are not readily available to other investigators.

(Remarks on code availability)

Version 1:

Reviewer comments:

Reviewer #1

(Remarks to the Author)

The authors have made substantial and constructive revisions. The statistical analysis was re-run using Welch's t-tests with a Holm-Bonferroni correction for multiple comparisons, directly addressing the main methodological concern from the first review round. The methods are now clearer, and the fundamental statistical flaws of the earlier version appear resolved.

The conclusions have shifted in response to the re-analysis. Rather than attributing performance differences to site bias, the revised results show few significant cross-site differences and emphasize cross-site robustness.

A remaining concern is that the comparison between MAE and DINOv2 is still confounded by differences in training protocols and optimization, which limits the strength of any conclusions about the relative performance of the two self-supervised learning (SSL) approaches.

The paper's main contribution is framed around its experimental design and the resulting empirical evidence. However, the design itself is not a novel contribution. It is a standard controlled experiment using off-the-shelf code on size-matched datasets. The subsequent evaluation is comprehensive and well executed.

The remaining weakness is the limited scope of the experiments. As the authors acknowledge (e.g., line 448), broader studies across learning strategies, task types, and medical domains with a replicated parallel design are desirable. In this work, the evidence is restricted to two large but specialized fundus image datasets, i.e., a single imaging modality. The observed cross-site robustness may be specific to the characteristics of the MEH and SDPP cohorts and may not extend to other fundus datasets, other retinal modalities (as hinted by the title) (for example OCT), or other areas of medical imaging.

Consequently, the claims about robustness of retinal foundation models to pre-training data shifts are not sufficiently supported by the current evidence beyond the tested data. Moreover, the observation that supervised fine-tuning can overcome domain gaps is well established; demonstrating it again in this narrow context does not constitute a significant advance.

(Remarks on code availability)

Reviewer #2

(Remarks to the Author)

All reviewer concerns have been satisfactorily addressed in the authors' extensive and complete revision and reply. This reviewer thanks the authors for the comprehensiveness of their revision and reply and for their consideration of queries posed.

(Remarks on code availability)

Version 2:

Reviewer comments:

Reviewer #3

(Remarks to the Author)

The authors' corrections substantially answer the concerns presented.

(Remarks on code availability)

Dear editors and reviewers,

Thank you for your valuable and instructive comments on our paper studying pre-training data effects on retinal foundation models. We have revised the manuscript in light of your suggestions to bring more insights. The major revisions are summarised below. All revisions in the manuscript are highlighted in blue. Please find our point-by-point response below.

Summary of major revisions to the manuscript

- Scope aligned to retinal FMs and images.
We have tightened the framing to retinal FMs and retinal images, removing statements that casually generalised to the broader medical domain. Revisions involve the Title, Abstract, Introduction, and Discussion sections.
- Statistical analysis strengthened.
We now report Welch's t-tests with Holm-Bonferroni correction. All raw p-values, adjusted p-values, and significance flags are provided in Supplementary Tables 3-8, and the relevant figures and main text have been updated accordingly.
- Expanded subgroup analysis.
The dataset for subgroup analysis has been increased from 2,000 images to 10,014 images to mitigate small-sample effects and improve robustness. Subgroup details are reported in Supplementary Table 4. The results in Figure 5, Supplementary Table 5, and the main text have been updated.
- Clarified research focus and key findings.
In the revision, we clarified that our primary focus is to quantify how pre-training data shape the generalisability and fairness of retinal FMs. Head-to-head comparisons between parallel retinal FMs indicate strong cross-site generalisability. Subgroup analyses reveal uneven impacts of pre-training demographic attributes on fairness. We removed text implying a practical data-selection framework (including original Figure 5a) from the Abstract, Introduction, and Discussion sections.
- Positioned within prior literature.
We added relevant references, particularly on AI model fairness and bias, to the Introduction and Discussion sections.
- Extra practical insight into model fairness.
We trained FM-MEH-SDPP using 300,000 synthetic images representative of the SDPP young cohort (age<40 years) and 604,170 real images from MEH. FM-MEH-SDPP achieved overall performance comparable to FM-MEH (Supplementary Table 7), with improved fairness in certain applications (Supplementary Table 8).

Reviewer #1

Paper Summary

This study investigates the influence of pre-training data characteristics (specifically, geographic origin and associated demographic distributions) on the generalizability and fairness of medical foundation models (FMs) in ophthalmology. The authors pre-train Vision Transformer-based FMs using two self-supervised learning methods (Masked Autoencoder (MAE), and DINOv2) on two large, distinct datasets of retinal fundus photographs: one from Moorfields Eye Hospital (MEH) in the UK and one from the Shanghai Diabetes Prevention Program (SDPP) in China. They evaluate these parallel-trained models (FM-MEH and FM-SDPP) on downstream tasks (diabetic retinopathy, diabetic macular edema, stroke prediction) using held-out data from both sites and several public datasets. Fairness is assessed by comparing performance across subgroups defined by age, sex, and ethnicity using MEH test data.

Main Claims and Observations

Fairness: The paper concludes that for these retinal FMs, the age distribution within the pre-training data introduces measurable performance bias across downstream task age subgroups, while pre-training data distributions for sex and ethnicity did not result in similar consistent biases. [lines 96, 269 following]

Generalizability & Site Bias: The study reports that FMs demonstrate good generalizability, performing comparably across sites and datasets in approximately 70% of evaluations. However, in the remaining ~30% of cases, FMs perform significantly better when the downstream task data aligns with their specific pre-training data source (intra-site advantage). [lines 322 following]

Response: Thank you very much for the accurate summary of our study. We highly appreciate your thoughtful comments and suggestions. We hope that our point-to-point response has addressed the concerns, and we are happy to provide further clarification.

Review Summary

1. This paper explores how pre-training data influences the behaviour of medical foundation models (FMs), employing a robust parallel training strategy on large-scale, high-quality datasets. While it addresses important questions, its main conclusions add little novelty or practical significance. The finding that models often perform better on data resembling their training distribution (site bias) is already a well-documented principle in machine learning.

Response: Thank you for acknowledging the robustness of our parallel training strategy and the significance of the research question. We agree that the impact of training data on machine learning models is well established, including the tendency for models to perform better on data aligned with their training distribution. Below, we clarify our specific contributions and their practical significance.

- According to our knowledge, this is the first study investigating FM generalisability and fairness using parallel retinal FMs. We developed two parallel retinal FMs, FM-MEH and FM-SDPP, that differ only in their pre-training data, to directly test how pre-training shapes generalisability and fairness on downstream tasks.
- Substantial and carefully controlled build. Constructing parallel FMs is non-trivial: we trained on two matched, large-scale, real-world datasets (~904k images each) from Moorfields Eye Hospital (MEH, UK) and the Shanghai Diabetes Prevention Program (SDPP, China), controlling computational resources and implementation details so that pre-training data were the only factor impacting FM performance.
- Why a head-to-head comparison between parallel FMs matters for generalisability. Prior work often compares FMs with baseline models on external data, which shows relative strength but does not investigate whether FMs are generalisable to address the domain gap challenge. Our comparisons between FM-MEH and FM-SDPP on both MEH and SDPP tasks provide direct evidence about this.
- Isolating the role of pre-training data in fairness. Using parallel FMs allows us to attribute fairness differences to pre-training data, rather than to model or training confounders, enabling clearer insight into demographic influences.
- Main finding 1: Strong cross-site generalisability. The parallel FMs performed comparably on most downstream tasks in each site, despite being pre-trained on different populations and imaging distributions. This partially contrasts with the conventional expectation that models preferentially perform on data resembling their training set, posing FMs as generalisable solutions for medical AI.

- Main finding 2: Demographic attributes have uneven fairness effects. We observed fairness gaps across age subgroups for retinal FMs, whereas sex and ethnicity did not show such effects. Data profile indicates substantial distributional variation (clinically meaningful morphological indices and latent features) across age groups, likely contributing to the observed age-related fairness gap.
- Practical mitigation via synthetic data. The age fairness improved when we supplemented pre-training data with synthetic images that provided a complementary age distribution.

In summary, our analyses provide robust evidence of how pre-training influences retinal FMs, highlighting the value of fine-grained, attribute-level data analysis and selection to optimise generalisability and fairness.

2. The approximately 30% figure reported here lacks deeper explanations of how or why this bias arises.

Response: Thank you for raising this concern. Before addressing this question, we first carefully revised the statistical analyses and added the corresponding description to the Methods and Results sections.

Line 617-623, Page 19:

“The normality of the model performance was checked via Shapiro-Wilk test. Statistical significance is calculated using Welch’s t-test, followed by Holm-Bonferroni correction. When comparing FM-MEH and FM-SDPP, the p-value was adjusted to the size of 24 (3*SDPP tasks, 3*MEH tasks, 6*publicly available datasets, each including fine-tuning and linear probing). In subgroup analysis, for each demographic attribute, the p-value was adjusted to the size of 8 (2*MEH tasks, 2*Self-supervised learning methods, each including fine-tuning and linear probing). All raw p-values, adjusted p-values, and significances are listed in Supplementary Table 3-8”

Line 215-220, Page 7:

“As shown in Figure 3, the FMs demonstrated good generalisability, with FM-MEH and FM-SDPP showing comparable performance on most evaluations. When adapted to SDPP downstream tasks (Figure 3a and 3b), FM-MEH and FM-SDPP performed comparably on all evaluations. Similarly, when adapted to MEH downstream tasks, FM-MEH significantly outperformed FM-SDPP on only three evaluations (ischaemic stroke prediction tasks), as shown in Figures 3c and 3d. AUPRC performance followed a similar trend, illustrated in Extended Data Figure 4.”

After strengthening the statistical analyses, we found that only three significant differences remained in comparing intra-site and inter-site adaptation performance (Figure 3), all relating to ischaemic stroke prediction at the MEH site. Potential reasons for this were discussed in the Discussion section.

Line 342-355, Page 10:

“Retinal FMs are more likely to demonstrate a generalisation gap in clinical tasks that hinge on subtle morphological manifestations, e.g. ischaemic stroke prediction in our study. FM-MEH significantly outperformed FM-Shanghai only for ischaemic stroke prediction, using both masked autoencoder and DINOv2 pre-training strategies (Figure 3). Compared to ocular disease detection (e.g. diabetic retinopathy and diabetic oedema), which targets well-defined lesions with abnormal colour and texture patterns, predicting systemic diseases from retinal images (termed as Oculomics [49,50]) typically depends on subtle changes in retinal anatomical tissues, such as vasculature and optic disc, arising from metabolic and circulatory dysfunction. Prior work has shown that retinal vascular calibre, tortuosity, fractal dimension, and optic-disc features carry systemic risk signals [40–46]. For example, narrower retinal arterioles and wider venules have been associated with long-term risk of mortality and ischaemic stroke (both sexes) and coronary heart disease (in women) [42], and patients with Alzheimer’s disease exhibit narrower venules with sparser, more tortuous vasculature compared with controls. Our results point to greater generalisation challenges for such Oculomics tasks and, for the time being, support using retinal FMs pre-trained on local data when available for these applications.”

Additionally, we investigated the generalisation gaps to publicly available datasets (Figure 4) and measured squared Maximum Mean Discrepancy to interpret the performance gaps.

Line 355-364, Page 11:

“Additionally, we measured the squared Maximum Mean Discrepancy (MMD²) between each public dataset and the MEH and SDPP cohorts (Supplementary Table 9) and visualised feature distributions using t-SNE (Extended Data Fig. 9). MMD² offers partial explanatory power for the generalisation gap between FM-MEH and FM-SDPP. For example, with MAE-based features, MESSIDOR2 distributes closer to SDPP than to MEH and FM-SDPP performs better; with DINOv2-based features, MESSIDOR2 is closer to MEH and FM-MEH performs better. However, this pattern does not hold for all datasets (e.g. Retina), indicating that distance metrics alone are insufficient to account for performance and generalisability differences. While FMs substantially mitigate cross-site generalisation gaps, residual disparities warrant further investigation.”

3. Furthermore, the finding that age distribution affects fairness, while potentially valid for this specific retinal imaging context, is presented with claims that overreach its generalizability to "medical FMs" broadly.

Response: Thank you for raising this issue. We have tightened the framing to retinal FMs and retinal images, removing statements that improperly generalised to the broader medical domain. These include:

Title: "Revealing the Impact of Pre-training Data on Retinal Foundation Models"

Abstract: "Medical foundation models (FMs), pre-trained on large-scale unlabelled data, show strong performance and data efficiency when adapted to various clinically relevant applications. However, how pre-training data shape the generalisability and fairness of these models, such as retinal FMs, remains unexplored. We address this using two large retinal image cohorts from Moorfields Eye Hospital (UK) and the Shanghai Diabetes Prevention Program (China), each containing 904,170 retinal images for FM pre-training. Using identical pipelines, we trained parallel FMs using individual image cohort and evaluated them on downstream tasks with publicly available datasets and held-out data from each site. The parallel FMs achieved comparable performance on most downstream tasks, indicating strong generalisability to data that differ substantially from their pre-training data. Nevertheless, we observed fairness gaps over age subgroups for retinal FMs, whereas sex and ethnicity showed no such impact. These results serve as quantitative evidence of the good generalisability of retinal FMs and reveal that pre-training demographic attributes shape fairness differently. This inspires domain-specific, fine-grained investigation of pre-training data effect to support evidence-based data selection to improve both generalisability and fairness in FM development."

Line 147-148, Page 5:

"To address these gaps, we investigated the impact of pre-training data on retinal FMs, one of the pioneering areas in medical FM"

Line 157-166, Page 5:

"Our results showed the strong generalisability of retinal FMs. For example, FM-MEH and FM-SDPP performed comparably across all SDPP downstream tasks. However, the models exhibited performance gaps over age subgroups (not seen in sex and ethnicity), likely caused by larger distributional shift across age subgroups. The age fairness improved when we supplemented pre-training data with synthetic images that provided a complementary age distribution. Through extensive experiments with real-world clinical data, this study examined the impact of pre-training data on retinal FMs. While our study focuses on retinal images, the observed combination of strong generalisability and uneven demographic effects may extend to broader medical

FMs. These findings suggest an evidence-based approach to data description and selection in FM development to improve both efficiency and capabilities.”

Line 315-321, Page 10:

“We developed parallel retinal FMs with identical implementations that differed only in their pre-training data, so as to isolate the pre-training data effect and provide robust evidence. Our results show that retinal FMs have good generalisability, achieving comparable performance in inter-site adaptation compared to intra-site one. Despite this, fairness gaps were observed across age subgroups, but not across sex or ethnicity, potentially driven by stronger age-related data variation captured by the FMs. These findings suggest fine-grained, attribute-level data analysis of pre-training data, working as quantitative evidence for efficient data selection in FM development.”

Line 365-366, Page 11:

“Pre-training data attributes, such as demographics, shape retinal FM generalisability and fairness differently, highlighting the need for fine-grained, attribute-level data analysis and selection.”

Line 373-378, Page 11:

“This suggests that, at least for retinal images, simply increasing range and balance of ethnicity and sex may not necessarily enhance FM fairness in downstream tasks, while ensuring a balanced and wide-ranging age distribution can improve both fairness and generalisability. These findings underscore the necessity of fine-grained data analysis for efficient data sampling in medical FM development, before uniformly scaling data across all metadata dimensions.”

Line 388-390, Page 12:

“In our retinal FM setting, age distribution produced significant subgroup differences in both morphological indices and latent features, and it significantly impacted the FM generalisability and fairness.”

Line 445-448, Page 13:

“Whether good generalisability exhibited by retinal FMs and uneven impact of data attributes extend to other task types (e.g. segmentation, regression tasks) and medical fields (e.g. pathology, radiology) is unknown. Further research involving a wider range of learning strategies, task types, and medical domains with a replicated parallel design is needed.”

Additionally, we replaced the term “medical FMs” with “retinal FMs” wherever it referred to our results and conclusions, to reflect the study’s scope accurately.

4. The finding regarding site-specific performance benefits is somewhat inconclusive, lacking a clear framework for practical data selection beyond reaffirming that data distribution matters.

Response: data shape the generalisability and fairness of retinal FMs. From extensive experiments, we draw two key findings: 1) retinal FMs generalise well across sites, and 2) demographic attributes influence fairness unevenly. These results partly depart from the conventional expectation that models perform best on data resembling their training distribution, motivating fine-grained, attribute-level data sampling in FM development to optimise both generalisability and fairness. While a practical data-selection framework based on these insights would be valuable, it lies beyond the scope of this work and warrants a dedicated study with tailored designs and comprehensive evaluation. Here, we address the core question, how pre-training data affect retinal FMs, by developing parallel retinal FMs, evaluating diverse downstream tasks, and conducting subgroup analyses across multiple demographic attributes. To avoid any confusion, we have removed language implying a practical data-selection framework (including the original Figure 5a) from the Abstract, Introduction, and Discussion sections.

5. Statistical methods require further clarification (especially given multiple comparisons), potentially undermining the reliability of the reported quantitative findings.

Response: This suggestion is really important and helpful. We now report Welch's t-tests with Holm-Bonferroni correction. The raw p-values, adjusted p-values, and significance flags are provided in Supplementary Tables 3-8, and the relevant figures and main text have been updated accordingly. This helps to provide a more reliable and robust statistical analysis. The revision includes:

Line 617-623, Page 19:

"The normality of the model performance was checked via Shapiro-Wilk test. Statistical significance is calculated using Welch's t-test, followed by Holm-Bonferroni correction. When comparing FM-MEH and FM-SDPP, the p-value was adjusted to the size of 24 (3*SDPP tasks, 3*MEH tasks, 6*publicly available datasets, each including fine-tuning and linear probing). In subgroup analysis, for each demographic attribute, the p-value was adjusted to the size of 8 (2*MEH tasks, 2*Self-supervised learning methods, each including fine-tuning and linear probing). All raw p-values, adjusted p-values, and significances are listed in Supplementary Table 3-8"

Line 215-220, Page 7:

“As shown in Figure 3, the FMs demonstrated good generalisability, with FM-MEH and FM-SDPP showing comparable performance on most evaluations. When adapted to SDPP downstream tasks (Figure 3a and 3b), FM-MEH and FM-SDPP performed comparably on all evaluations. Similarly, when adapted to MEH downstream tasks, FM-MEH significantly outperformed FM-SDPP on only three evaluations (ischaemic stroke prediction tasks), as shown in Figures 3c and 3d. AUPRC performance followed a similar trend, illustrated in Extended Data Figure 4.”

Line 226-235, Page 7:

“As shown in Figure 4, FM-SDPP and FM-MEH showed comparable performance in a majority of evaluations (8 out of 12 for Masked Autoencoder-based FMs; 10 out of 12 for DINOv2-based FMs). When pre-trained with Masked Autoencoder and fine-tuned to the downstream tasks (Figure 4a), FM-SDPP significantly outperformed FM-MEH on three out of six datasets (APTOS2019, MESSIDOR2, and Retina datasets), while the average performance gap on APTOS2019 is minor. When adapted to downstream tasks using linear probing (Figure 4c), FM-MEH performed significantly better on the Glaucoma fundus dataset ($p < 0.01$). When using DINOv2 for FM pre-training (Figure 4b and Figure 4d), FM-MEH significantly outperformed FM-SDPP in two applications, i.e. fine-tuning to MESSIDOR2 ($0.01 < p < 0.05$) and linear probing to IDRiD ($p < 0.01$).”

Line 240-253, Page 8:

“We evaluated FM fairness across demographic attribute subgroups in downstream tasks. Using a curated MEH dataset (details introduced in Supplementary Table 4), we examined subgroup performance on diabetic retinopathy detection and diabetic macular oedema detection, considering FM-SDPP and FM-MEH showed no significant differences in overall performance on these tasks (Figure 3c), and MEH data includes multiple ethnical groups. The SDPP pre-training data is mainly distributed over the young (<40 years) and middle-aged groups (40-70 years), with a subgroup ratio of (0.279, 0.685, 0.036), while MEH data is skewed towards middle-age and older cohorts (>70 years), with a ratio of (0.015, 0.467, 0.518). As shown in Figure 5a, FM-SDPP performed consistently better than FM-MEH in the young group (grey bars) but worse in the aged group (yellow bars). For instance, in diabetic retinopathy detection using a linear probe, FM-SDPP outperformed FM-MEH by an average AUROC of 0.073 in the young group ($p < 0.01$) while underperforming by 0.047 in the aged group ($p < 0.01$). FM-MEH and FM-SDPP demonstrated similar performance in the middle-aged group. This consistent performance gap across diverse applications demonstrated the FM bias introduced by differential age distribution in pre-training data.”

Revised Figures are listed below, while Supplementary Tables are not included here due to the table size:

Figure 3. AUROC performance of FM-MEH and FM-SDPP on downstream tasks using data from each site. Subgraphs **a** and **b** show the model performance on tasks at the SDPP site, with FMs respectively pre-trained with Masked Autoencoder and DINOv2. Subgraphs **c** and **d** present the performance of FMs on tasks at the MEH site. In SDPP downstream tasks, FM-SDPP and FM-MEH showed comparable performance, while FM-MEH achieved superior performance on 3 out of 12 evaluations when adapted to MEH downstream tasks. For each task, models were fine-tuned with five different random seeds, controlling the shuffling of fine-tuning data, and evaluated on the test set to generate five replicates. The mean AUROC values are represented by bar centres, with error bars indicating 95% confidence intervals (CI). A Welch's t-test followed by Holm-Bonferroni correction was used to assess whether the performance differences between FM-SDPP and FM-MEH were statistically significant. * indicates $0.01 < p < 0.05$ and ** indicates $p < 0.01$. *n* indicates the number of cases showing significant differences.

Figure 4. AUROC performance of FM-MEH and FM-SDPP on downstream tasks using publicly available datasets sourced from multiple countries. Subgraphs **a** and **b** show the performance of FMs pre-trained respectively with Masked Autoencoder and DINOv2 when fine-tuned to downstream tasks. Subgraphs **c** and **d** present the performance of FMs when adapted to downstream tasks with linear probing. When pre-trained with Masked Autoencoder, FM-SDPP significantly outperformed FM-MEH on 3 downstream evaluations. When pre-trained with DINOv2, FM-MEH significantly outperformed FM-SDPP on 2 evaluations. For each task, models were fine-tuned with five different random seeds, controlling the shuffling of fine-tuning data, and evaluated on the test set to generate five replicates. The mean AUROC values are represented by bar centres, with error bars indicating 95% CI. A Welch's t-test followed by Holm-Bonferroni correction was used to assess whether the performance differences between FM-SDPP and FM-MEH were statistically significant. * indicates $0.01 < p < 0.05$ and ** indicates $p < 0.01$. *n* indicates the number of cases showing significant differences.

Figure 5. Performance differences between FM-MEH and FM-SDPP on diabetic retinopathy and diabetic macular oedema detection across age (a), ethnicity (b), and sex (c) subgroups on MEH data. Yellow bars indicate that FM-MEH outperforms FM-SDPP, while grey bars indicate the opposite. The left two columns include the results for FMs pre-trained with Masked Autoencoder, while the right two columns show results for FMs pre-trained with DINOv2. Each section includes the results with fine-tuning and the linear probe. We observed consistent and significant performance gaps between FM-MEH and FM-SDPP in the age section, but not in the sex and ethnicity sections. The bar centres represent the mean AUROC differences. A Welch's t-test followed by Holm-Bonferroni correction was used to assess whether the performance differences between FM-SDPP and FM-MEH were statistically significant. * indicates $0.01 < p < 0.05$ and ** indicates $p < 0.01$. n indicates the number of cases showing significant differences.

Extended Data Figure 4. AUPRC performance of FM-MEH and FM-SDPP on downstream tasks using data from each site. Subgraphs **a** and **b** show the model performance on tasks at the SDPP site, with FMs respectively pre-trained with Masked Autoencoder and DINOv2. Subgraphs **c** and **d** present the performance of FMs on tasks at the MEH site. In SDPP downstream tasks, FM-SDPP and FM-MEH achieved comparable performance, while FM-MEH achieved superior performance on 4 out of 12 evaluations when adapted to MEH downstream tasks. For each task, models were fine-tuned with five different random seeds, controlling the shuffling of fine-tuning data, and evaluated on the test set to generate five replicates. The mean AUPRC values are represented by bar centres, with error bars indicating 95% CI. A Welch's t-test followed by Holm-Bonferroni correction was used to assess whether the performance differences between FM-SDPP and FM-MEH were statistically significant. * indicates $0.01 < p < 0.05$ and ** indicates $p < 0.01$. *n* indicates the number of cases showing significant differences.

Extended Data Figure 5. AUPRC performance of FM-MEH and FM-SDPP on downstream tasks using publicly available datasets sourced from multiple countries. Subgraphs **a** and **b** show the performance of FMs pre-trained respectively with Masked Autoencoder and DINOv2 when fine-tuned to downstream tasks. Subgraphs **c** and **d** present the performance of FMs when adapted to downstream tasks with the linear probe. When pre-trained with Masked Autoencoder, FM-SDPP significantly outperformed FM-MEH on 4 downstream evaluations. When pre-trained with DINOv2, FM-MEH significantly outperformed FM-SDPP on 2 evaluations. For each task, models were fine-tuned with five different random seeds, controlling the shuffling of fine-tuning data, and evaluated on the test set to generate five replicates. The mean AUROC values are represented by bar centres, with error bars indicating 95% CI. A Welch's t-test followed by Holm-Bonferroni correction was used to assess whether the performance differences between FM-SDPP and FM-MEH were statistically significant. * indicates $0.01 < p < 0.05$ and ** indicates $p < 0.01$. *n* indicates the number of cases showing significant differences.

6. The authors would benefit from more thorough contextualization of their results relative to previous works on fairness and bias in medical foundation models.

Response: Thank you for your suggestion. We have now added the description of previous literature in the Introduction and Discussion sections:

Line 131-133, Page 4:

“Prior studies have systematically shown that the distribution of fine-tuning data, particularly demographic attributes such as age and sex, can be encoded by AI models and lead to fairness gaps [23,25].”

Line 407-423, Page 12:

“This study provides quantitative evidence that the composition of pre-training data shapes the generalisability and fairness of retinal FMs. Some of our findings align with prior work on generalisability and fairness in medical imaging. For example, Yang et al. showed that models encoding more demographic attributes (e.g. age, sex) are prone to fairness gap under external evaluation across radiology, dermatology, and ophthalmology [60]. We extend this observation to FMs. As shown in Figure 5 and 6, differences in age distribution within the pre-training set are perceived by retinal FMs (age<40, 40<age<70, 70<age) and fairness gap across age subgroups is observed. Kerem et al. verified the good generalisability of vision FMs in medical image segmentation [24]. We also confirmed strong generalisability of retinal FMs. Importantly, rather than comparing FMs to baselines in external evaluation, we trained parallel FMs and conducted head-to-head intra-site vs inter-site comparison, showing that external performance matches internal performance, a stricter demonstration of generalisability. Wang et al. used auxiliary demographic tasks to improve external performance [25], while in our study, we verified that retinal FMs are highly generalisable. To our knowledge, this is the first study to use parallel retinal FMs to rigorously isolate the impact of pre-training data on both generalisability and fairness, an urgent and under-studied question. Together, these results both converge with the existing literature and advance it in FM area by showing how targeted pre-training curation can steer fairness improvement in medical foundation models.”

Line 425-434, Page 13:

“Our experiments with FM-MEH-SDPP echo prior findings that synthetic data can contribute effectively to FM pre-training [60,61]. Moreover, our fine-grained, age-conditioned image synthesis further verified the benefits of increasing the diversity and balance of age, a demographic attribute that strongly impacts both morphological indices and latent features. Although FM-MEH-SDPP performed comparably with FM-MEH on the general cohort, it significantly improved the performance on younger

cohorts on certain applications. While synthetic images cannot perfectly recapitulate real distributions, and we conditioned on age only, the observed subgroup improvements indicate that age is indeed an important metadata that should be prioritised in data selection for retinal foundation models. Nevertheless, signals on FM generalisability and how pre-training data impacts FM fairness may differ across medical domains and warrant domain-specific investigation, which we advocate through this study.”

7. Combined with numerous minor weaknesses, the paper overstates its contributions and requires substantial revision, potentially bordering on rejection due to the limited novelty and potential statistical issues.

Response: Thank you again for providing the detailed comments and suggestions. We hope our revision addressed your concerns.

Major Weaknesses

8. Limited Novelty and Depth of “Site-Bias” Finding. The core finding that models can exhibit better performance on data distributions matching their training data is not novel; it's a fundamental concept in domain adaptation and generalization research. While quantifying this effect for these specific FMs (~30% intra-site advantage) provides a data point, the paper falls short of providing significant new conceptual understanding or predictive insight into when or why this site bias occurs for FMs. This finding also stands in mild tension with literature suggesting large-scale self-supervised training can mitigate certain biases.

Response: Thank you for raising this concern. As noted in our response to Question 1, after revision, we report two main findings that we believe add value to the community:

- Main finding 1: Strong cross-site generalisability. The parallel FMs, FM-MEH and FM-SDPP, performed comparably on most downstream tasks in each site, despite being pre-trained on different populations and imaging distributions. This partially contrasts with the conventional expectation that models preferentially perform on data resembling their training set, posing FMs as generalisable solutions for medical AI.
- Main finding 2: Demographic attributes have uneven fairness effects. We observed fairness gaps across age subgroups for retinal FMs, whereas sex and ethnicity did not show such effects. Data profile indicates substantial distributional variation

(clinically meaningful morphological indices and latent features) across age groups, likely contributing to the observed age-related fairness gap.

We also noted that the three significant differences between FM-MEH and FM-SDPP at the MEH site pertain to ischaemic stroke prediction. We discuss potential reasons in the Discussion section

Line 341-363, Page 10-11:

“Retinal FMs are more likely to demonstrate a generalisation gap in clinical tasks that hinge on subtle morphological manifestations, e.g. ischaemic stroke prediction in our study. FM-MEH significantly outperformed FM-Shanghai only for ischaemic stroke prediction, using both masked autoencoder and DINOv2 pre-training strategies (Figure 3). Compared to ocular disease detection (e.g. diabetic retinopathy and diabetic oedema), which targets well-defined lesions with abnormal colour and texture patterns, predicting systemic diseases from retinal images (termed as Oculomics [49,50]) typically depends on subtle changes in retinal anatomical tissues, such as vasculature and optic disc, arising from metabolic and circulatory dysfunction. Prior work has shown that retinal vascular calibre, tortuosity, fractal dimension, and optic-disc features carry systemic risk signals [40–46]. For example, narrower retinal arterioles and wider venules have been associated with long-term risk of mortality and ischaemic stroke (both sexes) and coronary heart disease (in women) [42], and patients with Alzheimer’s disease exhibit narrower venules with sparser, more tortuous vasculature compared with controls. Our results point to greater generalisation challenges for such Oculomics tasks and, for the time being, support using retinal FMs pre-trained on local data when available for these applications. Additionally, we measured the squared Maximum Mean Discrepancy (MMD²) between each public dataset and the MEH and SDPP cohorts (Supplementary Table 9) and visualised feature distributions using t-SNE (Extended Data Fig. 9). MMD² offers partial explanatory power for the generalisation gap between FM-MEH and FM-SDPP. For example, with MAE-based features, MESSIDOR2 distributes closer to SDPP than to MEH and FM-SDPP performs better; with DINOv2-based features, MESSIDOR2 is closer to MEH and FM-MEH performs better. However, this pattern does not hold for all datasets (e.g. Retina), indicating that distance metrics alone are insufficient to account for performance and generalisability differences. While FMs substantially mitigate cross-site generalisation gaps, residual disparities warrant further investigation.”

Using parallel retinal FMs, our study provided robust evidence for FM generalisability and fairness:

- Why a head-to-head comparison between parallel FMs matters for generalisability. Prior work often compares FMs with baseline models on external data, which shows relative strength but does not investigate whether FMs are generalisable to address the domain gap challenge. Our comparisons between FM-MEH and FM-SDPP on both MEH and SDPP tasks provide direct evidence about this.
- Isolating the role of pre-training data in fairness. Using parallel FMs allows us to attribute fairness differences to pre-training data, rather than to model or training confounders, enabling clearer insight into demographic influences.

9. Concerns Regarding Statistical Rigor. Multiple Comparisons: The paper reports a large number of statistical tests across many tasks, datasets, models, and demographic subgroups (Figures 3, 4, 6, Supplementary Tables), yet there is no mention of adjusting for multiple comparisons (e.g., via Bonferroni correction). Reporting multiple unadjusted p-values raises the risk of overstating the significance of any observed differences, casting doubt on the reliability of the ~30% site-bias estimate and subgroup findings in Figure 6.

Response: Thank you. Please see our response to Question 5.

10. Subgroup Analysis & Sample Size: The authors draw fairness-related conclusions from subgroup analyses (Figure 6). Given the total dataset sizes and the age distribution data cited (Lines 281–282), sample counts within certain subgroups (especially ages <40 and >70) could be critically small, which may render results unstable or prone to outliers. The distribution of test samples and disease prevalence within these subgroups should be explicitly reported.

Response: We appreciate the reviewer highlighting this point. To minimise bias and sensitivity to outliers from small subgroup sizes, we expanded the subgroup analysis: the dataset increased from 2,000 to 10,014 images to improve analysis robustness. Detailed counts are provided in Supplementary Table 4. Even in the least represented age group (<40 years) at MEH site, case numbers for each disease category are reasonably adequate. Figure 5, Supplementary Table 5, and the corresponding main text have been updated accordingly.

Line 240-253, Page 8:

“We evaluated FM fairness across demographic attribute subgroups in downstream tasks. Using a curated MEH dataset (details introduced in Supplementary Table 4), we examined subgroup performance on diabetic retinopathy detection and diabetic macular oedema detection, considering FM-SDPP and FM-MEH showed no

significant differences in overall performance on these tasks (Figure 3c), and MEH data includes multiple ethnical groups. The SDPP pre-training data is mainly distributed over the young (<40 years) and middle-aged groups (40-70 years), with a subgroup ratio of (0.279, 0.685, 0.036), while MEH data is skewed towards middle-age and older cohorts (>70 years), with a ratio of (0.015, 0.467, 0.518). As shown in Figure 5a, FM-SDPP performed consistently better than FM-MEH in the young group (grey bars) but worse in the aged group (yellow bars). For instance, in diabetic retinopathy detection using a linear probe, FM-SDPP outperformed FM-MEH by an average AUROC of 0.073 in the young group ($p < 0.01$) while underperforming by 0.047 in the aged group ($p < 0.01$). FM-MEH and FM-SDPP demonstrated similar performance in the middle-aged group. This consistent performance gap across diverse applications demonstrated the FM bias introduced by differential age distribution in pre-training data.”

11. Prevalence: The potential influence of varying disease prevalence across demographic subgroups (e.g., DR/DME prevalence increasing with age) on subgroup performance metrics (like AUROC) needs clarification and discussion, even if the overall test sets were balanced. Especially in combination with small subgroup sizes <30 (age below 40 in MHE) low prevalence in this group may even result in missing samples for a class.

Response: Thank you for suggesting clarification on disease prevalence within subgroups. We have added fine-grained case counts for the subgroup analysis in Supplementary Table 4. MEH data, as a hospital-based cohort, exhibits generally high disease. For example, in the least represented age group (<40 years), the test set includes 21, 29, 31, 31, and 74 cases across the five stages of diabetic retinopathy, and 134, 26, and 26 cases for diabetic macular oedema. Accordingly, there is no long-tail issue or missing-class problem in the subgroup analysis.

12. Overgeneralization of Findings to “Medical Foundation Models”. Scope of “Medical FMs”: The paper repeatedly frames its findings in the context of “medical FMs” generally (Title, Abstract, Discussion). However, the empirical results are derived solely from 2D colour fundus photography, specific ViT architectures, and two SSL methods. The conclusion that age is the dominant fairness driver while sex/ethnicity are minimal is specific to this retinal context and cannot be readily generalized. The conclusion that age is the primary driver of fairness while sex and ethnicity show minimal effects applies to this retinal imaging context and cannot be generalized to all medical foundation models. Sex and ethnicity are known to be significant contributors to bias in other medical AI domains (e.g., radiology, dermatology), so any general statements require caution.

Response: Thank you. We agree that precise scoping is important. Please see our response to Question 3.

13. Limited Technical Scope: The results may not directly extend to different imaging modalities (CT, MRI), task types (segmentation, image generation), model architectures (CNNs, multimodal models), or other SSL/training methods.

Response: Thank you for raising this concern. While our results may not directly generalise to other imaging modalities, task types, or model architectures, our head-to-head evaluation of parallel retinal FMs, trained on two large, real-world cohorts using two representative self-supervised methods, offers valuable insights into how pre-training data influence retinal FM generalisability and fairness. These findings can inspire more efficient retinal FM development and, importantly, encourage domain-specific, fine-grained data analysis and selection across medical domains. We also acknowledge this as a limitation in the Discussion section:

Line 442-448, Page 13:

“Second, due to the considerable workload involved in developing parallel FMs and organising diverse downstream tasks, this study primarily focused on representative self-supervised learning strategies, such as Masked Autoencoder and DINOv2, and used retinal images as an exemplar. Whether good generalisability exhibited by retinal FMs and uneven impact of data attributes extend to other task types (e.g. segmentation, regression tasks) and medical fields (e.g. pathology, radiology) is unknown. Further research involving a wider range of learning strategies, task types, and medical domains with a replicated parallel design is needed.”

14. Claims of Offering a “Data Selection Framework”: The paper suggests its findings offer "practical guidance for improving the construction and application of medical FMs" (Lines 319-321) and proposes a "pipeline for key metadata identification" (Fig 5a, Lines 447-448). However, the practical advice boils down to selecting FMs pre-trained on similar data (Lines 339-340), which lacks operational specificity. The flowchart in Figure 5a adds little value; it's a linear description, not a decision-making tool. This falls short of providing a robust framework

Response: We apologise for the wording and resulting confusion. In the revision, we clarify that our primary aim is to quantify how pre-training data shape the generalisability and fairness of retinal FMs. Head-to-head comparisons between parallel retinal FMs show strong cross-site generalisability, and subgroup analyses reveal uneven impacts of pre-training demographic attributes on fairness. We have removed text implying a practical data-

selection framework (including the original Figure 5a) from the Abstract, Introduction, and Discussion.

15. Potentially Confounded SSL Comparisons. Comparing MAE and DINOv2 performance is complicated by vastly different reported pre-training protocols (epochs: 800 vs 100; batch size: 1792 vs 320) without discussion of hyperparameter tuning or ablation studies. Observed differences might stem from training budget or suboptimal parameters rather than solely the inherent properties of the SSL methods themselves for this task.

Response: We acknowledge that the previous descriptions of the MAE vs. DINOv2 comparison are arbitrary. Considering the complex confounders relevant to their performance, we explicitly introduced our experimental conditions, which do not necessarily produce the best-performing models for comparison.

Line 392-405, Page 12:

“Advancements in general-purpose AI techniques (e.g. self-supervised learning methods) continue to push the performance boundaries in our applications. In this study, we included and compared two representative self-supervised learning strategies, i.e. generative-based method (Masked Autoencoder) and self-distillation method (DINOv2). Under the same compute budget (identical GPU settings and training time), DINOv2 achieved superior performance in downstream tasks on both sites (Supplementary Table 5). This also extended to publicly available datasets (Supplementary Table 5). The advantage likely reflects a combination of pre-training strategies (i.e. DINO [58] and iBoT [59] capturing image-level and patch-level features respectively), practical refinements (e.g. Sinkhorn-Knopp centring [48]), and generalisable features learnt from initial large-scale pre-training on 142 million natural images [23]. However, it is noted that retinal FMs developed in this study, with either Masked Autoencoder or DINOv2, were not optimised to their theoretical best. We pre-trained these models out-of-the-box on large-scale retinal datasets, without extensive hyperparameter searches. Even so, these models achieved competitive performance across diverse downstream applications, illustrating the value of translational approaches that initialise medical FMs from evolving general AI models.”

Minor Weaknesses

16. Ambiguity in Figure 6 - The figure does not specify if the test data is from MEH, SDPP, or both combined, nor how the subgroups are distributed within those sets. Details on fine-tuning sets, class distributions, and data balancing are also unclear.

Response: Thank you for raising this issue. We have now explicitly mentioned that we used MEH data for subgroup analysis in the Figure caption and main text.

Figure 5. Performance differences between FM-MEH and FM-SDPP on diabetic retinopathy and diabetic macular oedema detection across age (a), ethnicity (b), and sex (c) subgroups on MEH data.

Yellow bars indicate that FM-MEH outperforms FM-SDPP, while grey bars indicate the opposite. The left two columns include the results for FMs pre-trained with Masked Autoencoder, while the right two columns show results for FMs pre-trained with DINOv2. Each section includes the results with fine-tuning and the linear probe. We observed consistent and significant performance gaps between FM-MEH and FM-SDPP in the age section, but not in the sex and ethnicity sections. The bar centres represent the mean AUROC differences. A Welch's t-test followed by Holm-Bonferroni correction was used to assess whether the performance differences between FM-SDPP and FM-MEH were statistically significant. * indicates $0.01 < p < 0.05$ and ** indicates $p < 0.01$. n indicates the number of cases showing significant differences.

Line 240-244, Page 8:

“We evaluated FM fairness across demographic attribute subgroups in downstream tasks. Using a curated MEH dataset (details introduced in Supplementary Table 4), we examined subgroup performance on diabetic retinopathy detection and diabetic macular oedema detection, considering FM-SDPP and FM-MEH showed no significant differences in overall performance on these tasks (Figure 3c), and MEH data includes multiple ethnic groups.”

17. Label Sources and Potential Biases - MEH retinal labels come from clinical records; SDPP labels come from expert annotations. This difference in label acquisition may partly underlie the performance differences, not just the site distribution. The paper briefly mentions it but does not delve into its implications.

Response: Thank you for the question. As we compared FM-MEH and FM-SDPP on MEH tasks and SDPP tasks separately, differences in label resources do not compromise fair comparisons. As shown in the study pipeline (Figure 1), the fine-tuning and evaluation data for each task were identical for both models, ensuring a fair comparison. We have also revised the Limitations paragraph to clarify this point

Line 448-453, Page 13:

“Third, due to the differences in the sources of labels (e.g. MEH diabetic retinopathy labels were extracted from clinical practice records; SDPP labels were annotated by two ophthalmologists with disagreements adjudicated by a consultant-level ophthalmologist), there are clear performance differences across various applications, as shown in Figure 3. Although this does not bias the performance comparison between FM-MEH and FM-SDPP, a well-aligned labelling system would enable extra cross-validation”

18. Unclear Link to Morphological Features - Although morphological feature differences between MEH and SDPP images are reported, the quantitative relationship between these differences and downstream task performance remains vague. The choice of morphological indices (e.g., fractal dimension) and their direct relevance to diabetic retinopathy, DME, or stroke prediction should be explored.

Response: Thank you for this question. We introduced the morphological indices for two purposes: 1) to characterise substantial differences between MEH and SDPP data as an additional modality of comparison (Figure 2 and the Results section), and 2) to quantify variation across demographic subgroups (Figure 6 and Extended Data Figures 6-7). While linking morphological indices (e.g. fractal dimension) to disease outcomes would give insights (explored in prior literature [40-46]), they may not directly contribute to our main findings.

19. Missing Pre-training Disease Distribution Details - The paper reports disease prevalence in the downstream sets but not in the pre-training sets. Figure 2 lacks information on the distribution of relevant diseases (DR, DME, stroke indicators if possible) within the MEH and SDPP pre-training datasets, which is highly relevant context. This information on the distribution of relevant diseases (DR, DME, and possibly stroke indicators) in these datasets should be included in Figure 2.

Response: Thank you for this helpful suggestion. We have added the disease prevalence in the main text and caption of Figure 2.

Line 184-187, Page 6:

“MEH data, as a hospital-based cohort, exhibits higher disease prevalence than SDPP data, which is derived from community screening. The prevalence of diabetic retinopathy, diabetic macular oedema, and ischaemic stroke in the MEH cohort is 8.71%, 3.02%, and 1.57%, respectively, compared with 1.4%, 0.08%, and 0.66% in the SDPP cohort.”

Line 874-876, Page 27:

“Regarding disease prevalence, MEH (hospital-based) shows higher disease prevalence than SDPP (community-based): diabetic retinopathy 8.71%, diabetic macular oedema 3.02%, and ischaemic stroke 1.57% in MEH vs 1.4%, 0.08%, and 0.66% in SDPP.”

20. Small Performance Gaps Treated as Significant - Some statistically significant differences reported involve very small absolute AUC changes (e.g., <0.01), questioning their practical relevance and the conclusion of one model outperforming the other. For

APTOS2019 AUC difference is 0.003; for Glaucoma fundus 0.006; for Diabetic retinopathy shanghai fine tune 0.006

Response: Thank you for raising this issue. After recalculating the p-values, most concerns have been resolved. For the remaining case, a minor but statistically significant difference between FM-MEH and FM-SDPP when fine-tuned to APTOS2019, we have added an explicit clarification in the main text.

Line 229-231, Page 7:

“When pre-trained with Masked Autoencoder and fine-tuned to the downstream tasks (Figure 4a), FM-SDPP significantly outperformed FM-MEH on three out of six datasets (APTOS2019, MESSIDOR2, and Retina datasets), while the average performance gap on APTOS2019 is minor”

21. Contradictory Results Across Architectures - In some cases, FM-SDPP outperforms FM-MEH for one SSL method (MAE) but performs worse under the other method (DINOv2) (e.g., MESSIDOR2, Retina datasets in Fig 4), making it hard to attribute performance solely to dataset differences versus interactions with the pre-training approach. Summing p-values across models is not appropriate.

Response: Thank you for this question. We have revised the main text to avoid summing p-values across models when evaluating and comparing FMs on publicly available datasets. The main text is:

Line 224-236, Page 7:

“We evaluated the generalisability of FM-MEH and FM-SDPP to diverse applications using six publicly available datasets, comprising diabetic retinopathy detection (APTOS2019, IDRiD, and MESSIDOR2), glaucoma detection (Glaucoma fundus), and multiple retinal disease detection (JSIEC and Retina). As shown in Figure 4, FM-SDPP and FM-MEH showed comparable performance in a majority of evaluations (8 out of 12 for Masked Autoencoder-based FMs; 10 out of 12 for DINOv2-based FMs). When pre-trained with Masked Autoencoder and fine-tuned to the downstream tasks (Figure 4a), FM-SDPP significantly outperformed FM-MEH on three out of six datasets (APTOS2019, MESSIDOR2, and Retina datasets), while the average performance gap on APTOS2019 is minor. When adapted to downstream tasks using linear probing (Figure 4c), FM-MEH performed significantly better on the Glaucoma fundus dataset ($p < 0.01$). When using DINOv2 for FM pre-training (Figure 4b and Figure 4d), FM-MEH significantly outperformed FM-SDPP in two applications, i.e. fine-tuning to MESSIDOR2 ($0.01 < p < 0.05$) and linear probing to IDRiD ($p < 0.01$).

AUPRC results are illustrated in Extended Data Figure 5. All quantitative results are listed in Supplementary Table 3.”

22. Figures and Flowcharts -The t-SNE clusterings mentioned in the text (Fig. 5b) are not clearly visible or explained. The flowchart in Figure 5a adds no value; it's a linear description, not a decision-making tool. Lack of discussion on metadata (age, sex, device, etc.) for the public datasets hinders interpretation of performance differences on these sets in the context of pre-training data similarity.

Response: Thank you for this feedback. In the revision, we clarify that our primary aim is to quantify how pre-training data shape the generalisability and fairness of retinal FMs. We removed language implying a practical data-selection framework (including the former Figure 5a) from the Abstract, Introduction, and Discussion. The updated Figure 6 (formerly Figure 5) now serves to illustrate variations in latent features and morphological indices across age subgroups.

We agree that discussing public dataset metadata (age, sex, device, etc.) would aid interpretation in the context of similarity to pre-training data. However, most public datasets provide limited demographic information, which constrains such analysis. To partly address this, we added a t-SNE visualisation to show latent-feature distributions of the public datasets with MEH and SDPP data in the Extended Data Figure 9. Additionally, we measured the squared Maximum Mean Discrepancy (MMD^2) between each public dataset and the MEH and SDPP cohorts (Supplementary Table 9), and added relevant discussion.

Extended Data Figure 9. t-SNE visualisation for MEH, SDPP data (5000 randomly sampled data points), and publicly available datasets used in this study, respectively with features extracted by foundation models developed in each site (FM-MEH and FM-SDPP) with Masked Autoencoder (MAE) and DINOv2.

Line 354-363, Page 11:

“Additionally, we measured the squared Maximum Mean Discrepancy (MMD^2) between each public dataset and the MEH and SDPP cohorts (Supplementary Table 9) and visualised feature distributions using t-SNE (Extended Data Fig. 9). MMD^2 offers partial explanatory power for the generalisation gap between FM-MEH and FM-SDPP. For example, with MAE-based features, MESSIDOR2 distributes closer to SDPP than to MEH and FM-SDPP performs better; with DINOv2-based features, MESSIDOR2 is closer to MEH and FM-MEH performs better. However, this pattern does not hold for all datasets (e.g. Retina), indicating that distance metrics alone are insufficient to account for performance and generalisability differences. While FMs substantially mitigate cross-site generalisation gaps, residual disparities warrant further investigation.”

Miscellaneous Issues

23. Precision Scores: AUPRC is calculated, but precision-recall trade-offs are not discussed in the text.

Response: Thank you for highlighting this. In this study, we constructed balanced downstream evaluation sets (equal numbers of cases and controls) to support fair model comparison and subgroup analyses. Given this design, we did not explore operating-point selection or precision–recall trade-offs, which are especially informative under imbalanced prevalence. We now acknowledge this as a limitation and, in future work, will 1) evaluate on datasets reflecting hospital- and community-level prevalence and 2) report operating-point analyses under realistic priors to better characterise precision–recall trade-offs.

Line 453-457, Page 13:

“Fourth, we balanced the downstream tasks by sampling equal numbers of disease and control images. This design facilitates fair model comparison and subgroup analyses. However, future evaluations should also include datasets that reflect community prevalence to assess real-world performance. Coupled with analysis at clinically relevant precision-recall operating points, this would provide deeper insight into performance trade-offs.”

24. Figure 2b Specificity: Mentions t-SNE for MAE features but not explicitly for DINOv2 features in the caption/text for comparison.

Response: Thanks, we have now included DINOv2 features in Extended Data Figure 1.

Extended Data Figure 1. t-SNE visualisation for MEH and SDPP data (5000 randomly sampled data points), respectively with features extracted by foundation models developed in each site (FM-MEH and FM-SDPP) with DINOv2.

25. Typo: Extended Data Figure 4 caption (line 900) likely means AUPRC, not AUROC, based on the figure content.

Response: Thank you for pointing out this mistake. We have corrected the typo.

26. Image Quality: Plotted in Fig 2c but its impact or relevance is not discussed.

Response: Thank you for this question. We included image quality (Figure 2c) to characterise substantial differences between MEH and SDPP images. For all downstream tasks at both sites, images underwent quality control with AutoMorph (as introduced in the Method section), minimising any bias in model performance caused by image quality.

27. Study Origins: The different clinical origins of the datasets (MEH/AlzEye vs. SDPP/Diabetes Prevention) might imply systematic differences beyond demographics (e.g., comorbidities, baseline health status) that are not explored.

Response: Thank you for this suggestion. We have added description in the main text. Although with the substantial differences between MEH and SDPP in data origins, demographics, morphological indices, and latent features distribution, retinal FMs achieved strong generalisability.

Line 184-187, Page 6:

“MEH data, as a hospital-based cohort, exhibits higher disease prevalence than SDPP data, which is derived from community screening. The prevalence of diabetic retinopathy, diabetic macular oedema, and ischaemic stroke in the MEH cohort is 8.71%, 3.02%, and 1.57%, respectively, compared with 1.4%, 0.08%, and 0.66% in the SDPP cohort.”

28. Age Binning: Justification for binning age rather than analyzing it as a continuous variable or using more granular bins is missing.

Response: Thanks, we have revised Figure 2a to show granular age bins (one year per bar) for the MEH and SDPP cohorts.

Strengths:

29. Parallel Training Design: The core strength is training identical FM architectures with identical methods (per SSL type) on two distinct, large-scale datasets, providing a clean setup to isolate the impact of pre-training data origin.

Valuable Datasets: Access to and use of large, real-world clinical datasets (MEH, SDPP) adds significant value.

Important Problem: The study addresses a timely and critical question about the foundations of generalizability and fairness in medical FMs.

Response: Thank you for recognising our parallel-training design. We appreciate your acknowledgement of the value of large-scale datasets and the importance of our investigation into FM generalisability and fairness.

Suggestions

30. Deeper Bias Analysis: Investigate potential mechanisms behind observed biases. How do models encode ethnicity/gender, even if downstream performance differences weren't consistently significant in this study?

Response: Thank you for this suggestion. We streamlined the paper's logic as follows: we first developed parallel retinal FMs and evaluated them across multiple sites and downstream tasks; we then assessed fairness across demographic subgroups and identified fairness gaps by age; next, we explored why an age-related gap emerged while gaps by sex and ethnicity did not. We examined data variation (latent features and morphological indices) across demographic groups and found substantial differences by age, which likely contribute to the observed fairness gap. Visualising latent features with t-SNE (Figure 6, Extended Data Figure 6-7) provides an alternative view of how models encode demographic attributes. Our findings align with prior literature.

Line 409-413, Page 12:

“For example, Yang et al. showed that models encoding more demographic attributes (e.g. age, sex) are prone to fairness gap under external evaluation across radiology, dermatology, and ophthalmology [60]. We extend this observation to FMs. As shown in Figure 5 and 6, differences in age distribution within the pre-training set are perceived by retinal FMs (age<40, 40<age<70, 70<age) and fairness gap across age subgroups is observed.”

Line 271-293, Page 8-9:

“We probed the reasons why demographic attribute unevenly shaped the model performance, particularly fairness over subgroups. We mainly investigated this from two aspects, the morphological indices and latent features across demographic subgroups in pre-training data. As shown in Figure 6, we observed distinct distribution differences across age subgroups in both latent features (Figure 6a and 6c) and morphological indices (Figure 6e and 6f). After controlling for confounders by looking at only Asian or Asian British females (Figure 6b and 6d), there are still significant gaps in morphological indices (Figure 6f). This demonstrates that age subgroups exhibit clear morphological variations, and even with self-supervised pre-training, retinal FMs learned distinct latent features across age subgroups, which likely caused bias in FM fairness.

For ethnicity, we investigated three subgroups White, Asian or Asian British, and Black or Black British. The t-SNE visualisations (Extended Data Figure 6a, FMs trained with Masked Autoencoder) showed clear clustering partially for the White cohort. When FMs were pre-trained with DINOv2 (Extended Data Figure 6c), latent features showed no distinct clustering across all ethnicities. Additionally, when controlling for confounders, there were no significant differences across ethnic subgroups in morphological indices (Extended Data Figure 6f).

For sex subgroups (i.e. female, male), we observed no distinct clustering in t-SNE visualisations, either before or after removing confounding variables (Extended Data Figure 7). Only the vein fractal dimension showed significant differences across the sex subgroups after removing confounding variables (Extended Data Figure 7f). These findings suggest that, ethnicity and sex subgroups contributed limited observable variations in morphological indices and less distinct latent compared to age distribution, which are less likely to bias FM fairness and generalisability in downstream tasks.”

31. Combined Dataset Training: Explore if pre-training on a combined MEH+SDPP dataset mitigates the observed age bias and site-specific performance gaps.

Response: Thank you for this suggestion. We agree that pre-training on a combined dataset could yield additional insights, but merging real-world clinical data is challenging due to privacy regulations. As a workaround, we synthesised images representative of the SDPP

cohort by training a generative model on SDPP images and conditioning synthesis on age. Using these synthetic images to complement the pre-training age distribution improved fairness across age subgroups in certain applications (Supplementary Table 8). We have added details in the Results and Discussion sections.

Line 295-309, Page 9:

“Inspired by the observations above, we tested whether retinal FM generalisability and fairness improve when pre-training on a dataset with a more diverse and balanced age distribution. We generated 300K synthetic colour fundus photographs representative of SDPP young cohort (age<40) in age distribution, using synthetic models (details introduced in the Method section) and merged them with randomly sampled 604,170 MEH images, for a total of 904,170 images. Image synthesis was explicitly conditioned on age to match the age profile in SDPP young cohort. The age distribution of this combined data is shown in Extended Data Figure 8. Using this combined dataset, we trained FM-MEH-SDPP using DINOv2 and compared it with FM-MEH in subgroup analysis. As shown in Supplementary Table 7, FM-MEH-SDPP performed comparably with FM-MEH, with differences observed for diabetic oedema detection on MEH data and diabetic retinopathy detection on MESSIDOR2 datasets after linear probing. For model fairness across age subgroups (Supplementary Table 8), FM-MEH-SDPP showed significant improvement in young groups for diabetic retinopathy detection after fine-tuning ($p=0.02$ after Holm-Bonferroni correction) and for diabetic retinopathy after linear probing ($p=0.033$ after correction). These results indicate that simply increasing age distribution diversity and balance in the pre-training data can yield fairness gains, even with synthetic images.”

Line 425-434, Page 13:

“Our experiments with FM-MEH-SDPP echo prior findings that synthetic data can contribute effectively to FM pre-training [60,61]. Moreover, our fine-grained, age-conditioned image synthesis further verified the benefits of increasing the diversity and balance of age, a demographic attribute that strongly impacts both morphological indices and latent features. Although FM-MEH-SDPP performed comparably with FM-MEH on the general cohort, it significantly improved the performance on younger cohorts on certain applications. While synthetic images cannot perfectly recapitulate real distributions, and we conditioned on age only, the observed subgroup improvements indicate that age is indeed an important metadata that should be prioritised in data selection for retinal foundation models. Nevertheless, signals on FM generalisability and how pre-training data impacts FM fairness may differ across medical domains and warrant domain-specific investigation, which we advocate through this study.”

Extended Data Figure 8. Age distribution of the combined dataset, comprising 604,170 real images from MEH and 300,000 synthetic images representative of the younger SDPP cohort (age < 40 years). Pre-training the retinal FM on this combined dataset allows us to examine whether age-related fairness improves, particularly within the younger subgroup in downstream tasks. We note that image synthesis may not always strictly follow the age condition. However, as the SDPP data used for Stable Diffusion XL training are drawn from a young cohort (age<40 years), the resulting synthetic images should still be broadly representative of this group.

32. SSL Comparison Rigor: Provide justification for the chosen training protocols or conduct ablation studies to better isolate the effect of the SSL strategy itself versus training budget/hyperparameters, especially for claims of DINOv2 superiority.

Response: Thank you for this suggestion. We have enhanced the description rigour. More details are included in the response to Question 15.

33. Expand Scope (Future Work): Acknowledge limitations more strongly and suggest future work replicating this parallel design in other medical domains (e.g., radiology, pathology) to assess the generalizability of these specific findings.

Response: Thank you for this suggestion. We have narrowed the description to retinal FMs and retinal images, as detailed in our response to Question 3. We also stated this explicitly in the limitations paragraph.

Line 442-448, Page 13:

“Second, due to the considerable workload involved in developing parallel FMs and organising diverse downstream tasks, this study primarily focused on representative self-supervised learning strategies, such as Masked Autoencoder and DINOv2, and

used retinal images as an exemplar. Whether good generalisability exhibited by retinal FMs and uneven impact of data attributes extend to other task types (e.g. segmentation, regression tasks) and medical fields (e.g. pathology, radiology) is unknown. Further research involving a wider range of learning strategies, task types, and medical domains with a replicated parallel design is needed”

34. Statistical Correction: Re-evaluate statistical significance using appropriate corrections for multiple comparisons and report corrected p-values or use confidence intervals more centrally. Clearly report subgroup sample sizes and prevalence.

Response: We highly appreciate this valuable suggestion. We have updated all statistical analyses. Please see our response to Question 5.

35. Contextualize Novelty: Reframe the "site bias" finding by clearly acknowledging it confirms existing principles and focus novelty on the quantification within the FM/retinal context, or on deeper analysis if performed.

Response: Thank you for this suggestion. We have included prior literature as context. Please see our response to Question 6.

Reviewer #2

Noteworthy results

Yukun Zhou and co-authors, in “Revealing the Impact of Pre-training Data on Medical Foundation Models,” explore fairness and generalizability of medical foundation models, using ophthalmic fundus photography as the prototypical case.

The most important finding is the impact of certain metadata on foundation model (FM) generalizability, in particular with regard to age. Since the authors created two different foundation models, one with MEH data and another with SDPP data. The groups differed across a few domains, most notably age. The investigators demonstrated differences between foundation model performance when comparing MEH and SDPP analysis of datasets and also showed the effect of age within models. This is a key finding, as it speaks to the importance of transparency and bias in model fairness and generalizability with regard to the datasets used to create the foundation models.

Ethnicity and sex did not appear to affect model performance. The investigators showed “that FMs demonstrate[d] good generalizability, achieving comparable performance in over 70% of downstream tasks. However, FMs sometimes perform[ed] better on application data that aligns with their pre-training data.”

The authors point out that “long-term and sustainable advancement of medical AI requires the development of domain-specific techniques tailored to the unique characteristics of medical data and application scenarios... Pre-training clinical data and self-supervised learning strategies have a synergistic effect on FM performance... need to simultaneously optimise both model learning strategies and pre-training data characteristics to advance medical FM development.”

The authors make several excellent and important recommendations for future work in this area, including the following:

- “Future studies should include extra factors particularly concerning disease phenotypes.”
- “Further research involving a wider range of learning strategies and medical domains is needed.”
- “There are clear performance differences across various applications.”

Significance to the field. This work has significant potential to enhance the quality of FMs used for analysis of medical data. It is an interesting examination of the role and effect of training data on FM performance and bias.

Conclusions and Claims

The authors’ conclusions and claims are supported by their data.

Flaws in the data analysis, interpretation and conclusions

No significant flaws are noted.

Response: Thank you very much for the thoughtful and generous appraisal of our work. We appreciate your clear summary, particularly the emphasis on demographic effects, cross-site generalisability, the variable impacts of ethnicity and sex, and the recommendations for future work. We hope this study encourages fine-grained, attribute-level data analysis and selection across medical domains to support efficient FM development.

1. Of note is that the MEH and SDPP samples used to create and test the FMs were unbalanced and a subset of data were used to create balanced and curated samples. The authors should comment on how this might have affected the FMs and comparisons.

The methodology is sound, well thought out and well executed. As noted above, the authors should comment on the imbalance of the MEH and SDPP datasets.

Response: Thank you for this helpful suggestion. The MEH (hospital-based) and SDPP (community-based) cohorts are indeed imbalanced in disease prevalence (e.g. diabetic retinopathy, diabetic macular oedema, and ischaemic stroke). To enable fair model comparison and subgroup analyses, we constructed balanced downstream evaluation sets with equal numbers of disease cases and controls. While this design reduces issue of data imbalance and supports stable estimation of group-wise effects, it also departs from real-world prevalence, which can affect the evaluation on external applicability. We now acknowledge this as a limitation and, as part of future work, will 1) evaluate on datasets reflecting hospital- and community-level prevalence, and 2) report operating-point analyses under realistic priors to provide deeper insight into performance trade-off.

Line 184-187, Page 6:

“MEH data, as a hospital-based cohort, exhibits higher disease prevalence than SDPP data, which is derived from community screening. The prevalence of diabetic retinopathy, diabetic macular oedema, and ischaemic stroke in the MEH cohort is 8.71%, 3.02%, and 1.57%, respectively, compared with 1.4%, 0.08%, and 0.66% in the SDPP cohort.”

Line 453-457, Page 13:

“Fourth, we balanced the downstream tasks by sampling equal numbers of disease and control images. This design facilitates fair model comparison and subgroup analyses. However, future evaluations should also include datasets that reflect community prevalence to assess real-world performance. Coupled with analysis at

clinically relevant precision-recall operating points, this would provide deeper insight into performance trade-offs.”

3. The authors have provided adequate information to enable reproduction of their results, although their data are not readily available to other investigators.

Response: Thank you for your comments on reproducibility. We provide implementation details to replicate our study and have released all code and model weights (see the Code availability section). Public dataset links are listed, and access to private clinical data follows the standardised application process outlined in the Data availability section.

Dear editors and reviewers,

Thank you for acknowledging our previous efforts in addressing the concerns and improving the manuscript. We have now added further revision and clarification to address the remaining concerns. All revisions are marked in blue in the manuscript. Please find our point-by-point response below.

Reviewer #1

1. The authors have made substantial and constructive revisions. The statistical analysis was re-run using Welch's t-tests with a Holm-Bonferroni correction for multiple comparisons, directly addressing the main methodological concern from the first review round. The methods are now clearer, and the fundamental statistical flaws of the earlier version appear resolved.

The conclusions have shifted in response to the re-analysis. Rather than attributing performance differences to site bias, the revised results show few significant cross-site differences and emphasize cross-site robustness.

Response: Thank you for appreciating our substantial revisions. We are pleased that the main concerns have been addressed. The updated analyses and results demonstrate strong cross-site generalisation of the trained foundation models, comparable to intra-site adaptation on most clinical applications.

2. A remaining concern is that the comparison between MAE and DINOv2 is still confounded by differences in training protocols and optimization, which limits the strength of any conclusions about the relative performance of the two self-supervised learning (SSL) approaches.

Response: Thank you for this insightful comment. We acknowledge that our previous discussion may cause some confusion. We intended to compare RETFound-MAE and RETFound-DINOv2, rather than to directly contrast the self-supervised learning (SSL) strategies, MAE and DINOv2, in isolation. Even in the original DINOv2 paper [48], it is challenging to conclude the intrinsic superiority of one SSL strategy over another, given that the compared models were trained with distinct data and optimisation protocols. In our study, we aimed to compare RETFound-MAE and RETFound-DINOv2 using identical pre-training retinal images and computational budgets, but differing in their initial weights and optimisation approaches, across diverse downstream applications. To clarify this point, we have revised the relevant discussion as follows:

Line 393-406, Page 12:

“Advancements in general-purpose AI methods continue to push the performance boundaries in our applications. In this study, we incorporated two representative self-

supervised learning paradigms, a generative-based method (Masked Autoencoder [47]) and a self-distillation-based method (DINOv2 [48]), to develop RETFound-MAE and RETFound-DINOv2, using the same compute budget (identical GPU settings and training duration). Our experiments showed that RETFound-DINOv2 achieved superior performance in downstream tasks across both sites (Supplementary Table 5), and this trend extended to publicly available datasets. This advantage likely reflects that strong initial weights from general-purpose AI models, such as DINOv2, can facilitate subsequent domain-specific foundation model development and applications. However, it should be noted that the retinal foundation models developed in this study, RETFound-MAE and RETFound-DINOv2, were not optimised to their theoretical best. Both were pre-trained out-of-the-box on large-scale retinal datasets without extensive hyperparameter tuning. Even so, they achieved competitive performance across diverse downstream applications, highlighting the translational potential of medical foundation models initialised from evolving general AI models.”

3. The paper's main contribution is framed around its experimental design and the resulting empirical evidence. However, the design itself is not a novel contribution. It is a standard controlled experiment using off-the-shelf code on size-matched datasets. The subsequent evaluation is comprehensive and well executed.

Response: Thank you for acknowledging that our evaluation is comprehensive and well executed. We would like to clarify that we have not claimed novelty in our experimental design, either in the manuscript or in the response. Our study is not a typical methodological development work that introduces new algorithms or methods. Instead, its main contribution lies in providing robust empirical evidence on how pre-training data impacts the generalisability and fairness of retinal foundation models through extensive experiments. The insights are derived from a carefully controlled and standardised experimental setting.

We believe this evidence provides substantial value to the broader research community, including those involved in both the development and deployment of foundation models. For instance, the observed strong generalisability of retinal foundation models may encourage researchers to prioritise the collection of large-scale datasets from new imaging modalities (e.g. OCTA) to develop complementary foundation models, rather than replicating datasets for existing modalities, such as retinal fundus photographs, for local pre-training. Likewise, those applying retinal foundation models for clinical tasks may consider demographic alignment, such as age distribution, more critical than geographic origin. Given the rapid expansion of retinal foundation model research and their wide use across diverse applications [1–5], we believe these findings are both highly relevant and impactful.

In addition, we have made our model weights fully open-source, together with detailed usage instructions (<https://github.com/rmaphoh/RETFound>). This provides another strong retinal

foundation to advance research and applications in foundation models, ocular disease detection, and oculosics tasks.

4. The remaining weakness is the limited scope of the experiments. As the authors acknowledge (e.g., line 448), broader studies across learning strategies, task types, and medical domains with a replicated parallel design are desirable. In this work, the evidence is restricted to two large but specialized fundus image datasets, i.e., a single imaging modality. The observed cross-site robustness may be specific to the characteristics of the MEH and SDPP cohorts and may not extend to other fundus datasets, other retinal modalities (as hinted by the title) (for example OCT), or other areas of medical imaging.

Response: We agree that expanding the scope of experiments across broader learning strategies, task types, and medical domains is desirable. This is exactly why we acknowledged it as one of the limitations and directions for future work. However, to the best of our knowledge, this study represents the first quantitative investigation into how pre-training data impacts the generalisability and fairness of retinal foundation models in clinical applications, with multiple parallel foundation models and extensive evaluation.

The primary bottleneck in starting this line of research lies in the substantial experimental demands, developing multiple foundation models and conducting downstream evaluations and subgroup analysis, while carefully isolating the effects of pre-training data. **Our current study already reflects a significant scale of investigation.** Specifically, we included:

- Two large-scale cohorts (~904K retinal fundus photographs each)
- Two representative learning strategies (MAE and DINOv2);
- Five foundation models (RETFound-MAE and RETFound-DINOv2 trained on each site separately, plus one trained on merged data, including real MEH data and synthetic data representative of the SDPP cohort);
- Evaluation across six downstream datasets from two sites, covering both ocular disease detection and oculosics tasks (diabetic retinopathy, diabetic oedema, and ischaemic stroke prediction);
- Six publicly available datasets representing different ocular diseases and varied demographics;
- A large-scale fairness analysis across demographic subgroups; and
- Two adaptation strategies (fine-tuning and linear probing).

These experiments collectively underpin six main figures, nine Extended Data Figures, and nine Supplementary Tables (certain Figures and Tables include massive amounts of data). While extending this study to additional modalities (e.g., OCT or OCTA) would be valuable, it is beyond the feasible scope of a single research article, also requiring substantial

computational and data availability (e.g. 904K OCT or OCTA scans from two large-scale, distinct cohorts for pre-training; multiple curated datasets for evaluation).

We view this work as an initial but important step toward a comprehensive and evidence-based understanding of foundation model generalisability and fairness in medical imaging. Although it cannot cover all possible learning strategies or imaging modalities, as no single study can, it provides solid empirical evidence and aims to encourage future collective efforts from the research community to advance this critical area.

That said, we appreciate the reviewer's suggestion for clearer framing. While we had specified the use of retinal fundus photographs in the introduction (e.g. line 150, page 5) and the following sections (e.g. line 172, page 6), we have now revised the title and abstract to explicitly highlight this focus, ensuring clarity and precision for the audience.

Title: Revealing the Impact of Pre-training Data on Retinal Foundation Models: Evidence from Two Large-Scale Fundus Photograph Cohorts

Abstract: Medical foundation models (FMs), pre-trained on large-scale unlabelled data, show strong performance and data efficiency when adapted to various clinically relevant applications. However, how pre-training data shape the generalisability and fairness of these models, such as retinal FMs, remains unexplored. We address this using two large retinal **fundus photograph** cohorts from Moorfields Eye Hospital (UK) and the Shanghai Diabetes Prevention Program (China), each containing 904,170 retinal **fundus photographs** for FM pre-training. Using identical pipelines, we trained parallel FMs using individual image cohort and evaluated them on downstream tasks with publicly available datasets and held-out data from each site. The parallel FMs achieved comparable performance on most downstream tasks, indicating strong generalisability to data that differ substantially from their pre-training data. Nevertheless, we observed fairness gaps over age subgroups for retinal FMs, whereas sex and ethnicity showed no such impact. These results serve as quantitative evidence of the good generalisability of these FMs and reveal that pre-training demographic attributes shape fairness differently. This inspires domain-specific, fine-grained investigation of pre-training data effect to support evidence-based data selection to improve both generalisability and fairness in FM development.

5. Consequently, the claims about robustness of retinal foundation models to pre-training data shifts are not sufficiently supported by the current evidence beyond the tested data. Moreover, the observation that supervised fine-tuning can overcome domain gaps is well established; demonstrating it again in this narrow context does not constitute a significant advance.

Response: Thank you for sharing your detailed and constructive feedback in this second-round review. We hope that the revisions and clarifications provided above have addressed your concerns regarding the study scope.

We agree that fine-tuning is a well-established method for mitigating domain gaps. However, this is not the central focus of our work. Instead, our findings go beyond the expected benefits of fine-tuning: through extensive experiments, we observed that retinal foundation models demonstrate strong cross-site generalisability, achieving performance comparable to intra-site adaptation using a retinal foundation model pre-trained on the same site. This reflects not only a narrowing of performance gaps, but also an impressive level of generalisability rarely observed in broader machine learning models.

Furthermore, our linear probing analyses independently confirmed this high level of generalisation. Beyond generalisability, our study also provides insights into fairness, identifying disparities across demographic subgroups and demonstrating practical strategies for partially improving fairness by involving synthetic data to pre-train retinal foundation models.

Overall, we believe this work makes a meaningful contribution to the field by providing rigorous empirical evidence on the generalisability and fairness of retinal foundation models. Our previous RETFound model has made significant contributions to the ophthalmic AI community, serving as one of the most widely used open-source models for diverse clinical and research applications [1-5]. In this study, we further introduce RETFound-DINOv2, a new, generalisable, and openly available retinal foundation model that we expect will continue to benefit a broad range of ocular disease detection and oculomics studies. Additionally, these research findings, particularly regarding generalisation and fairness, will help optimise future foundation model development and deployment efforts across the wider community, as detailed in the response to Question 3.

References:

- [1] Sun, Yuqi, et al. "A data-efficient strategy for building high-performing medical foundation models." *Nature Biomedical Engineering* (2025): 1-13.
- [2] Wang, Meng, et al. "Enhancing diagnostic accuracy in rare and common fundus diseases with a knowledge-rich vision-language model." *Nature Communications* 16.1 (2025): 5528.
- [3] Jiang, Nan, et al. "A deep learning system for detecting silent brain infarction and predicting stroke risk." *Nature Biomedical Engineering* (2025): 1-13.
- [4] Lee, Tae Kwan, et al. "Vision transformer based interpretable metabolic syndrome classification using retinal Images." *npj Digital Medicine* 8.1 (2025): 205.
- [5] Zhang, Juzhao, et al. "RETFound-enhanced community-based fundus disease screening: real-world evidence and decision curve analysis." *NPJ digital medicine* 7.1 (2024): 108.

Open-source Models:

<https://github.com/rmaphoh/RETFound>

Reviewer #2

1. All reviewer concerns have been satisfactorily addressed in the authors' extensive and complete revision and reply. This reviewer thanks the authors for the comprehensiveness of their revision and reply and for their consideration of queries posed.

Response: Thank you for acknowledging that the concerns have been fully addressed. Your insightful comments have greatly helped us improve the manuscript.